# Chain of Thought Empowers Transformers to Solve Inherently Serial Problems

**Zhiyuan Li**
TTIC & Stanford University
zhiyuanli@ttic.edu

**Hong Liu**
Stanford University
hliu99@stanford.edu

**Denny Zhou**
Google DeepMind
dennyzhou@google.com

**Tengyu Ma**
Stanford University
tengyuma@stanford.edu

## Abstract

Instructing the model to generate a sequence of intermediate steps, *a.k.a.*, a chain of thought (CoT), is a highly effective method to improve the accuracy of large language models (LLMs) on arithmetics and symbolic reasoning tasks. However, the mechanism behind CoT remains unclear. This work provides a theoretical understanding of the power of CoT for decoder-only transformers through the lens of expressiveness. Conceptually, CoT empowers the model with the ability to perform inherently serial computation, which is otherwise lacking in transformers, especially when depth is low. Given input length $n$, previous works have shown that constant-depth transformers with finite precision $\text{poly}(n)$ embedding size can only solve problems in $\mathsf{TC}^0$ without CoT. We first show an even tighter expressiveness upper bound for constant-depth transformers with constant-bit precision, which can only solve problems in $\mathsf{AC}^0$, a proper subset of $\mathsf{TC}^0$. However, with $T$ steps of CoT, constant-depth transformers using constant-bit precision and $O(\log n)$ embedding size can solve any problem solvable by boolean circuits of size $T$. Empirically, enabling CoT dramatically improves the accuracy for tasks that are hard for parallel computation, including the composition of permutation groups, iterated squaring, and circuit value problems, especially for low-depth transformers.

## 1 Introduction

Large Language Models (LLMs) exhibit exceptional capabilities in complex reasoning tasks such as mathematical problem-solving and code generation (Chowdhery et al., 2023; Anil et al., 2023; Achiam et al., 2023; Romera-Paredes et al., 2023; Trinh et al., 2024), far surpassing standard supervised machine learning techniques. The key to unlocking these advanced reasoning abilities lies in enabling LLMs to generate intermediate steps, or a chain of thought (CoT), before finalizing the final answer. This can be achieved through various methods, including training or instruction tuning a model with examples enriched with intermediate steps (Ling et al., 2017; Cobbe et al., 2021; Nye et al., 2021; Chung et al., 2022), or through zero-shot or few-shot CoT prompting (Reynolds & McDonell, 2021; Nye et al., 2021; Wei et al., 2022; Kojima et al., 2022).

A natural explanation is that the intermediate steps provide extra information about the tasks and efficient approaches to solving, so that a model can imitate. However, intriguingly, the efficacy of generating thought steps extends to zero-shot CoT prompting (Kojima et al., 2022), where LLMs are only instructed with the prompt "let's think step by step", and to even using incorrect reasoning steps in the few-shot examples (Wang et al., 2022a; Madaan & Yazdanbakhsh, 2022). These observations suggest that the form of few-shot CoT prompting is as important as (if not more important than) its content, because merely instructing LLMs to generate the intermediate steps helps.

This paper aims to study why the form of CoT improves the reasoning capability of LLMs. Our hypothesis is that CoT allows for performing more serial computations that a vanilla transformer cannot do without CoT. We formulate and analyze this hypothesis through the lens of expressiveness with and without CoT. We adopt the language of circuit complexity to discuss the capability

of transformers. Previous works (Liu et al., 2022b; Merrill & Sabharwal, 2023b) have shown standard decoder-only transformers (that output answers directly) are efficient parallel computers and can only express functions computable in an $O(1)$-parallel run-time with threshold circuits, $\mathsf{TC}^0$, a computational model that allows the AND, OR, NOT and MAJORITY function with multiple inputs to be computed efficiently in parallel. We first show a tighter upper bound (Theorem 3.1) for expressiveness of constant-precision transformer – it can only express a proper subset class of $\mathsf{TC}^0$, $\mathsf{AC}^0$, where MAJORITY gates are not allowed. Our upper bound is also more realistic because it handles the rounding issue or iterative addition of floating point numbers, while most previous results essentially only work for fixed-point number addition.

We then show that transformers equipped with CoT—allowing the transformer to auto-regressively generate a sequence of intermediate tokens before answering the questions—can solve complex problems that inherently require serial computations (assuming well-known conjectures in complexity theory). Intuitively, without CoT, the number of serial computations conducted by the transformer is bounded by the depth (which is considered as a fixed constant for this work), whereas with $T$ intermediate steps, the number of serial computations possible is boosted to $T$. Note that $T$ can easily increase as the sequence length increases where the depth is a fixed number that depends on the architecture.

Concretely, we prove that a constant-precision transformer with $T$ intermediate steps and embedding dimension logarithmic in the sequence length can express any functions computable by a circuit of size $T$ in Theorem 3.3. Taking $T$ to be polynomial in the sequence length, the result suggests that transformers with polynomially many intermediate steps are capable of computing all circuits in with polynomial size, P/poly, a superclass of P. Theorem 3.3 also implies that transformers with linearly many intermediate steps can compute all regular languages, including composition of non-solvable groups, like permutation group over five elements, $S_5$, which does not belong to $\mathsf{AC}^0$ and is also widely conjectured to be out of $\mathsf{TC}^0$. As such, CoT makes transformers with bounded depth and precision strictly more powerful.

To corroborate our theoretical analysis, we empirically evaluate the capability of transformers in solving four core problems: modular addition, permutation composition, iterated squaring, and circuit value problem. We train transformers to solve these tasks with a large amount of synthetic data, with and without CoT, or with additional hint but not CoT. The modular addition belongs to $\mathsf{TC}^0$, meaning it can be easily solved in parallel. Liu et al. (2022a) shows it is solvable by constant-depth transformers with log-precision and, indeed empirically depth 1 is sufficient for the parity problem (Modulo 2 addition). The other three tasks are all conjectured to require inherently serial computations. As expected, the vanilla transformer either requires a huge depth to solve these tasks (because the depth is the upper bound on the number of serial computation by transformers), or cannot solve the tasks at all. On the other hand, CoT can solve these tasks as long as the depth exceeds a small threshold. These experiments demonstrate CoT can provide more serial computations to solve complex reasoning tasks.

## 2 NOTATIONS AND PRELIMINARIES

Given a string $x$ or a set $x$, we use $|x|$ to denote the size of $x$. For any $n \in \mathbb{N}^+$, we define $\mathrm{softmax} : \mathbb{R}^n \to \mathbb{R}^n$ as $(\mathrm{softmax}(x))_i = \exp(x_i)/\sum_{i=1}^n \exp(x_i)$ for any $x \in \mathbb{R}^n$ and $i \in [n]$. We refer the readers to Appendix B for all notations and preliminaries on circuit complexity and transformers.

Given a vocabulary $\mathcal{V}$, a *decoder-only* transformer with parameter $\theta$ and maximal input length $n_{\max}$ maps a sequence of input tokens $(x_1, \ldots, x_n) \in \mathcal{V}^n$ to a probability distribution over $\mathcal{V}$ for all $n \le n_{\max}$, denoted by $p_\theta(\cdot \mid x_1, \ldots, x_n)$. We also define function $\mathsf{TF}_\theta(x)$ by the token in $\mathcal{V}$ that maximizes $p_\theta(\cdot \mid x_1, \ldots, x_n)$, that is, $\mathsf{TF}_\theta(x) \triangleq \arg\max_{x \in \mathcal{V}} p_\theta(x \mid x_1, \ldots, x_n)$.

**Next-token Generator:** Given a vocabulary $\mathcal{V}$, a next-token generator with parameter $\theta$ and maximal input length $n_{\max}$ is a mapping from $\cup_{n=1}^{n_{\max}} \mathcal{V}^n$ to $\mathcal{V}$. The main next-token generator we are interested in this work is decoder-only transformers, $\mathsf{TF}_\theta(x_1, \ldots, x_n)$ where $x_i \in \mathcal{V}$ for all $i \in [n]$. We also recursively define $\mathsf{TF}_\theta^i(x_1, \ldots, x_n) \triangleq \mathsf{TF}_\theta^{i-1}(x_1, \ldots, x_n, \mathsf{TF}_\theta(x_1, \ldots, x_n))$, for every positive integer $i$ and $n$ satisfying that $i + n \le n_{\max} - 1$ with the base case that $\mathsf{TF}_\theta^1(x_1, \ldots, x_n) \triangleq \mathsf{TF}_\theta(x_1, \ldots, x_n)$. In other words, for all $0 \le i \le n_{\max} - n - 1$, the output with $i$ steps of CoT is $x_{n+i+1} = \mathsf{TF}_\theta^{i+1}(x_1, \ldots, x_n) = \mathsf{TF}_\theta(x_1, \ldots, x_n, x_{n+1}, \ldots, x_{n+i})$.

**Transformer Architecture Overview:** The decoder-only transformer model we consider in this paper is very similar to GPT style architectures (Radford et al., 2019) and consists of four parts: a token embedding layer (TE), a position encoding layer (PE), an output linear layer (OUTPUT), and a stack of $L$ identical layers serving as the "decoder" where $L$ is also called the depth of the model. Each decoder layer has two sub-layers: a multi-head self-attention layer (ATTN) and a position-wise fully-connected feed-forward network (FF). Each layer mentioned above has its own trainable parameters and is indexed by the layer name and the depth for attention and feedforward layers. [1] That is we can split the model parameter $\theta$ in the following way: $\theta = (\theta_{\mathsf{PE}}, \theta_{\mathsf{TE}}, \theta_{\mathsf{OUTPUT}}, \{\theta^{(l)}_{\mathsf{ATTN}}, \theta^{(l)}_{\mathsf{FF}}\}_{l=0}^{L-1})$, which are all trainable. (See formal definition in Algorithm 1). We defer the details of the standard transformer layers, including the token embedding layer TE, the positional encoding layer PE, the attention layer ATTN, the feedforward layer FF, and the output layer OUTPUT into Appendix B.2.

---

**Algorithm 1** Decoder-only Transformer, $\mathsf{TF}_\theta$ and $p_\theta$

---

**Input:** Transformer parameter $\theta = (\theta_{\mathsf{PE}}, \theta_{\mathsf{TE}}, \theta_{\mathsf{OUTPUT}}, \{\theta^{(l)}_{\mathsf{ATTN}}, \theta^{(l)}_{\mathsf{FF}}\}_{l=0}^{L-1})$ and input tokens $x = (x_1, \ldots, x_n) \in \mathcal{V}^n$.
**Output:** Output distribution $p_\theta(\cdot \mid x_1, \ldots, x_i)$ for all $i \in [n]$ and output token $\mathsf{TF}_\theta(x)$.
1: $h_i^{(0)} \leftarrow \theta_{\mathsf{TE}}(x_i) + \theta_{\mathsf{PE}}(i), \forall i \in [n]$
2: **for** $l = 0, \ldots, L-1$ **do**
3: $\quad (h_1^{(l+0.5)}, \ldots, h_n^{(l+0.5)}) \leftarrow (h_1^{(l)}, \ldots, h_n^{(l)}) + \mathsf{ATTN}_{\theta^{(l)}_{\mathsf{ATTN}}}(h_1^{(l)}, \ldots, h_n^{(l)})$
4: $\quad h_i^{(l+1)} \leftarrow h_i^{(l+0.5)} + \mathsf{FF}_{\theta^{(l)}_{\mathsf{FF}}}(h_i^{(l+0.5)}), \forall i \in [n]$
5: **end for**
6: $p_\theta(\cdot \mid x_1, \ldots, x_i) \leftarrow \mathsf{OUTPUT}_{\theta_{\mathsf{OUTPUT}}}(h_i^{(L)}), \forall i \in [n]$
7: $\mathsf{TF}_\theta(x) \leftarrow \arg\max_x p_\theta(x \mid x_1, \ldots, x_n)$.

---

# 3 EXPRESSIVENESS THEORY FOR TRANSFORMERS WITH CHAIN OF THOUGHT(COT)

## 3.1 FINITE PRECISION MODELING

In practice, training and inference of transformers are typically done with 16- or 32-bit floating point numbers. Thus in this paper, we mainly focus on the computation model of *constant-precision* transformers, where the output of each arithmetic operation is rounded to the closest floating point number representable by a fixed number of digits following IEEE 754 standard (Definition B.2), thus avoiding the unrealistic infinite precision assumption made by prior works (Pérez et al., 2019; Dehghani et al., 2018).

Below we give an informal definition of finite-precision transformers and defer the formal definition to Appendix B.3. We use $\mathbb{F}_{e,s}$ to denote the set of floating-point numbers that can be expressed by 1 sign bit, $e$ exponent bit, and $2s$ precision bits. We further use $\mathsf{round}_{e,s}(x)$ to denote the closest number to $x$ in $\mathbb{F}_{e,s}$, which is also denoted by $[\cdot]_{e,s}$ for convenience. We break the tie by picking the one with a smaller absolute value. We extend the definition of $\mathsf{round}_{e,s}$ to vector inputs by rounding coordinate-wisely.

Next, we define finite-precision summation over more two numbers by decomposing it as a chain of rounded binary addition in a fixed order. [2]

**Definition 3.1** (Summation with Iterative Rounding). For any $s, n \in \mathbb{N}^+$ and vector $x \in \mathbb{R}^n$, we define *summation with iterative rounding to $2s$-bit precision* as $\mathsf{sum}_{e,s} : \cup_{n \in \mathbb{N}^+}(\mathbb{F}_{e,s})^n \to \mathbb{F}_{e,s}$, where

for any $n \in \mathbb{N}^+$ and $x \in \mathbb{R}^n$, $\mathsf{sum}_{e,s}(x) \triangleq \left[\left[\left[[x_1 + x_2]_{e,s} + x_3\right]_{e,s} + \cdots x_{n-1}\right]_{e,s} + x_n\right]_{e,s}$.

---

[1] We ignore the LayerNorm (Ba et al., 2016) in the usual transformer architecture for simplicity. Our expressiveness analysis can extend to the transformers with LayerNorm with more careful treatment. See Appendix G.1 for discussion.

[2] Technically speaking, instead of a chain, the summation could also proceed like a tree. This is a more complicated case and we leave it for future work.

We further define the following operations:

- Finite-precision inner product: $\langle x, y \rangle_{e,s} \triangleq \mathsf{sum}_{e,s}(x \odot y)$;
- Finite-precision matrix product: $(A \times_{e,s} B)_{i,j} \triangleq \left\langle (A_{i,:})^\top, B_{:,j} \right\rangle_{e,s}$;
- Finite-precision softmax: $\mathrm{softmax}_{e,s}(x) \triangleq \left[ [\exp(x)]_{e,s} / \mathsf{sum}_{e,s}([\exp(x)]_{e,s}) \right]_{e,s}$.

Finally, a finite-precision transformer can be defined by replacing all the infinite-precision operations by their finite-precision counterparts listed above. (See details in Algorithm 4). We postpone the details of the finite-precision version of individual transformer layers into Appendix C.

## 3.2 CoT: Complexity Class for Constant-depth Transformers with CoT

In this subsection, we define the complexity class consisting of all the problems that can be solved by some decoder-only transformers with CoT with finite precision.

**Definition 3.2** (CoT). Given a finite vocabulary $\mathcal{V}$ and four functions $T(n), d(n), s(n), e(n)$, informally, $\mathsf{CoT}[T(n), d(n), s(n), e(n)]$ is the family of problems solvable by a transformer with a constant depth, $s(n)$ bits of precision, $e(n)$ bits of exponent, embedding size $d(n)$ and $T(n)$ steps of CoT. Formally, we say a problem $\mathcal{L} : \cup_{n \in \mathbb{N}^+} \mathcal{V}^n \to \{0, 1\}$ is in $\mathsf{CoT}[T(n), d(n), s(n), e(n)]$ iff there is an integer $L$ and three functions $T'(n) = O(T(n)), d'(n) = O(d(n)), s'(n) = O(s(n))$, $e'(n) = O(e(n))$, such that for every positive integer $n$, there is a $L$-layer decoder-only transformer, denoted by $\mathsf{TF}_{\theta_n}$ with embedding size $d'(n)$, $2s'(n)$ bits of precision, and $e'(n)$ bits of exponent, that can output $\mathcal{L}(x)$ given any input $x$ in $\mathcal{V}^n$, using $T'(n)$ steps of chain of thought. Mathematically, it means

$$\mathsf{TF}_{\theta_n}^{1+T'(n)}(x) = \mathcal{L}(x), \quad \forall x \in \mathcal{V}^n. \tag{1}$$

We also extend the definition of CoT to a class of function instead of a single function. For example, $\mathsf{CoT}[T(n), \mathsf{poly}(n), s(n), e(n)] \triangleq \cup_{k \in \mathbb{N}^+} \mathsf{CoT}[T(n), n^k, s(n), e(n)]$.

**Definition 3.3** (T). We define $\mathsf{T}[d(n), s(n)] \triangleq \mathsf{CoT}[0, d(n), s(n)]$ as the problems that a constant-depth, constant-precision decoder-only transformer can solve with $O(s(n))$ bits of precision, $O(d(n))$ embedding size and without CoT (or with only 0 step of CoT).

By definition, $\mathsf{CoT}[T(n), d(n), s(n), e(n)]$ is monotone in all $T(n), d(n), s(n), e(n)$, *e.g.*, $\mathsf{CoT}[T'(n), d(n), s(n), e(n)] \subseteq \mathsf{CoT}[T(n), d(n), s(n), e(n)]$ if $T'(n) \leq T(n)$ for all $n \in \mathbb{N}$. In particular, we have $\mathsf{T}[d(n), s(n)] \triangleq \mathsf{CoT}[0, d(n), s(n)] \subseteq \mathsf{CoT}[T(n), d(n), s(n)]$.

Note the above-defined complexity class CoT is non-uniform, that is, it allows a different program for every input size. This is in contrast to previous works (Pérez et al., 2019; 2021; Yao et al., 2021; Weiss et al., 2021; Chiang et al., 2023; Hao et al., 2022; Merrill & Sabharwal, 2023a; Merrill et al., 2022) which focus on the uniform transformer classes. Please refer to Appendix H for a discussion.

## 3.3 Tighter Upper Bounds on Transformer Expressiveness

Existing works (Merrill & Sabharwal, 2023b; Liu et al., 2022a) have shown that constant depth, polynomial width, and log precision transformers can be simulated in a small parallel time, *i.e.*, using $\mathsf{TC}^0$ circuits. These results are built on the fact that multiplication and division of $n$-bits binary numbers (Hesse, 2001), as well as the iterated addition over $n$ different $n$-bit binary integers are in $\mathsf{TC}^0$.

However, such $\mathsf{TC}^0$ expressiveness upper bounds may be unrealistic for transformers operating with floating point numbers. (Merrill & Sabharwal, 2023b; Liu et al., 2022a) implicitly assumes when adding more than one floating-point numbers, the algorithm first compute the exact answer without rounding using arbitrarily more precision and only perform rounding in the end. However, in practice rounding happens after each addition between two numbers and it is open if such $\mathsf{TC}^0$ upper bounds still holds. Immediate rounding makes iterated addition over floating point numbers no longer associative (Goldberg, 1991), for example, $\mathsf{round}(a + \mathsf{round}(b + c)) \neq \mathsf{round}(\mathsf{round}(a + b) + c)$. The associativity of integer addition plays a crucial role in the fact that the iterated addition over $n$ different $n$-bit binary integers are in $\mathsf{TC}^0$.

In this section, we present two novel expressiveness upper bounds for transformers which rounds the immediate result after step of arithmetic operation. First, we show a strictly tighter upper bound

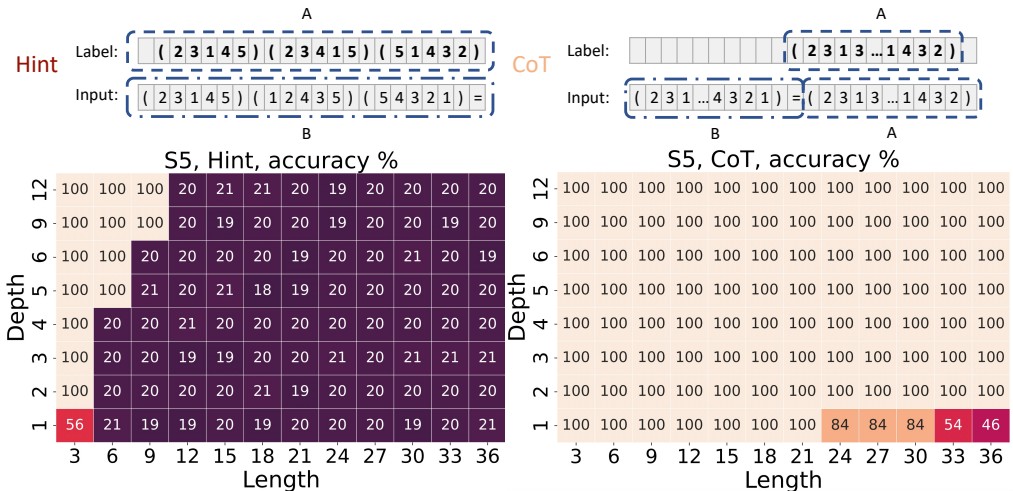

**Figure 1:** Permutation Composition ($S_5$). The label is the composition of all the permutations, where given two permutation $\sigma = (\sigma_1, \ldots, \sigma_5)$, $\pi = (\pi_1, \ldots, \pi_5)$, we define $\sigma \circ \pi \triangleq (\sigma_{\pi_1}, \ldots, \sigma_{\pi_5})$. The chain of thoughts and hints are the partial compositions. Only CoT can solve this task well, as predicted by our Theorem 3.5. Note for the most time the accuracy without CoT is $\sim 20\%$, which is no better than randomly guessing a number between 1 and 5.

than $\mathsf{TC}^0$, which is $\mathsf{AC}^0$, for constant-depth transformers with both constant bits of precision and exponents. (Theorem 3.1) This suggests when input length is sufficiently long, constant-precision transformers cannot count eventually, even in the sense of modular. For example, it is well known that no $\mathsf{AC}^0$ circuits can decide the parity of a binary string.

**Theorem 3.1.** $\mathsf{T}[\mathrm{poly}(n), 1, 1] \subseteq \mathsf{CoT}[1, \mathrm{poly}(n), 1, 1] \subseteq \mathsf{AC}^0$.

Our second result, Theorem 3.2, shows that when the number of bits for exponent is 0 (*i.e.* fixed-point numbers), $\mathsf{TC}^0$ upper bounds for expressiveness of constant-depth, log-precision transformers still holds, even with the correct rounding defined in Definition B.2.

**Theorem 3.2.** $\mathsf{T}[\mathrm{poly}(n), \log(n), 0] \subseteq \mathsf{CoT}[1, \mathrm{poly}(n), \log(n), 0] \subseteq \mathsf{TC}^0$.

The main technical difficulties in above two results are showing $\mathrm{sum}_{e,s} : (\mathbb{F}_{e,s})^n \to \mathbb{F}_{e,s}$ has $\mathsf{AC}^0$ (resp. $\mathsf{TC}^0$) circuits when $e, s$ are both constants (resp. $e = 0$, $s = O(\log(n))$). We view iterated addition with rounding over $\mathbb{F}_{e,s}$ as an automaton with both state space and vocabulary being $\mathbb{F}_{e,s}$. The first result are due to a novel application of classical Krhon-Rhodes decomposition theorem for automata (Theorem D.2), where we use the property of rounded addition that for all $x, x' \in \mathbb{F}_{e,s}, y \in \mathbb{F}_{e,s}, x \geq x' \implies [x + y]_{e,s} \geq [x' + y]_{e,s}$. We formalize this property in Definition E.2 as *ordered* automata and show all ordered automata are counter-free Theorem E.3 and thus can be simulated by $\mathsf{AC}^0$ circuits (McNaughton & Papert, 1971).

The proof technique for Theorem 3.1 does not generalize to Theorem 3.2 because the depth of $\mathsf{AC}^0$ circuits constructed before depends on the number of the states of the automaton and thus is not constant. Our proof for Theorem 3.2 is motivated by Algorithm 1 in Liu et al. (2022a) for the automaton named 'GridWorld'.

However, it remains open whether constant-depth, log-precision transformers with log bits for exponents $\mathsf{T}[\mathrm{poly}(n), \log(n), \log(n)]$ or even constant bits for exponents $\mathsf{T}[\mathrm{poly}(n), \log(n), 1]$ have $\mathsf{TC}^0$ circuits.

### 3.4 CoT Makes Transformers More Expressive

Now we are ready to present our main theoretical results (Theorem 3.3) which characterize the expressiveness of constant-depth, constant-precision transformers with CoT and $O(\log(n))$ embedding size. $\log(n)$ embedding sizes is necessary to ensure that the position embeddings for $n$ inputs are different. All the lower bounds for transformer expressiveness (with or without CoT) are proved for fixed-point numbers, *i.e.*, without using any exponent bits. Allowing exponent bits will only make transformers more expressive. For convenience, we define $\mathsf{CoT}[T(n), d(n), s(n)] \triangleq \mathsf{CoT}[T(n), d(n), s(n), 0]$. The omitted proofs in this section can be found in Appendix F.

**Theorem 3.3.** For any polynomial function $T : \mathbb{N}^+ \to \mathbb{N}^+$, $\mathsf{SIZE}[T(n)] \subseteq \mathsf{CoT}[T(n), \log n, 1]$. In particular, $\mathsf{P/poly} = \mathsf{CoT}[\mathsf{poly}(n), \log n, 1]$.

Compared to Theorems 3.1 and 3.2, Theorem 3.3 shows that allowing polynomial steps of CoT strictly makes constant-depth, constant-precision, decoder-only transformer more expressive and log-precision transformers more expressive under a standard hardness assumption that $\mathsf{TC}^0 \subsetneq \mathsf{P/poly}$.[3]

*Proof sketch of Theorem 3.3.* The high-level proof idea is that we use each step in CoT to simulate one gate operation in the target circuit and write the gate output as next input. To do that, we use one position encoding to store the information for each gate, which contains four parts: the current gate type $\{\mathsf{AND}, \mathsf{OR}, \mathsf{NOT}, \mathsf{TRUE}, \mathsf{FALSE}\}$, the two input gates id and the current gate id. Since there are total $\mathsf{poly}(n)$ gates, $d(n) = \Theta(\log n)$ embedding size suffices to store the above information. And the CoT here is constructed to be the values of each gate in the increasing order of id. Therefore, in each step, we can use attention to pull the value (either computed already or it is input) of the two input gates and use a feedforward network to compute the value of the current gate. □

As we can see from proof sketch, a crucial step for CoT to simulate any depth circuit is to write the output token back to the next input position. This action resets the "depth" of the intermediate output in the circuit to $0$. Our theory explains the ablation experiment in Wei et al. (2022) that when the model is prompted to output a only sequence of dots (. . .) equal to the number of tokens needed to solve the problem, the performance is no better than directly outputting the answer.

Because every regular languages can be recognized by a finite state automaton (Definition D.1) and finite state automata can clearly be simulated by linear size circuits. The following holds as a direct corollary of Theorem 3.3

**Corollary 3.4.** Every regular language belongs to $\mathsf{CoT}[n, \log n, 1]$.

Below we give a concrete regular language that constant-depth, poly-embedding-size transformers can solve only with CoT, the wording problem of permutation group over 5 elements, $S_5$ in Theorem 3.5, under a standard hardness assumption that $\mathsf{TC}^0 \subsetneq \mathsf{NC}^1$ (Yao, 1989).

**Definition 3.4** (Wording problem of group $G$)**.** Given $n$ elements from $G$, $(g_1, \ldots, g_n)$, we use $\mathcal{L}_G$ to denote the decision problem of whether $g_1 \circ g_2 \circ \cdots \circ g_n$ is equal to the identity of $G$.

For convenience, in this paper we extend the domain of $\mathcal{L}_G$ to the sequence of groups encoded by binary strings. The proof of Theorem 3.5 is a direct consequence of Theorems 3.2, 3.3 and 3.6.

**Theorem 3.5.** Assuming $\mathsf{TC}^0 \subsetneq \mathsf{NC}^1$, the wording problem of $S_5$ , $\mathcal{L}_{S_5}$ is in $\mathsf{CoT}[n, \log n, 1]$ but not $\mathsf{T}[\mathsf{poly}(n), \log n]$.

**Theorem 3.6** (Barrington (1986))**.** The wording problem of $S_5$ is $\mathsf{NC}^1$-complete under $\mathsf{AC}^0$ reductions. That is, for any decision problem $\mathcal{L}$ in $\mathsf{NC}^1$, there is a family of $\mathsf{AC}^0$ circuits $\{C_n\}_{n=1}^\infty$ (constant depth, $\mathsf{poly}(n)$ fan-outs), such that for any $n \in \mathbb{N}^+$ and $x \in \{0, 1\}^n$, $\mathcal{L}(x) = \mathcal{L}_{S_5}(C_n(x))$.

*Proof of Theorem 3.5.* First $\mathcal{L}_{S_5}$ is a regular language, thus belonging to $\mathsf{CoT}[n, \log n, 1]$ by Corollary 3.4. Since $\mathcal{L}_{S_5}$ is $\mathsf{NC}^1$-complete by Theorem 3.6, assuming $\mathsf{TC}^0 \subsetneq \mathsf{NC}^1$, $\mathcal{L}_{S_5}$ does not belong to $\mathsf{TC}^0$. This proof is completed by applying Theorem 3.2, which says $\mathsf{T}[\mathsf{poly}(n), \log(n)] \subseteq \mathsf{TC}^0$. □

**Results for** $\mathsf{poly}(n)$ **embedding size:** So far we have been focusing on the expressiveness of transformer with $O(\log n)$ embedding size, so it is natural to ask can whether transformers can also benefit from having a larger embedding size, say $\mathsf{poly}(n)$? Our Theorem 3.7 answers this question positively by showing that log-precision (resp. constant-precision) constant-depth poly-embedding-size decoder-only transformers with $T(n)$ steps of CoT can simulate any $T(n)$-size circuit with some $\mathsf{TC}^0$ (resp. $\mathsf{AC}^0$) oracle gates with $\mathsf{poly}(n)$ input.

Formally, given a decision problem $\mathcal{L} : \cup_{n=1}^\infty \{0, 1\}^n \to \{0, 1\}$, we use $\mathcal{L}_n$ to denote the restriction of $\mathcal{L}$ on $\{0, 1\}^n$, which can also be viewed as an single gate with $n$ fan-ins. We define problems that can be solved by circuits with certain size of gates (including oracle gates) by Definition 3.5. [4]

---

[3]Indeed such separation can be shown for *any* polynomial steps of CoT by padding polynomially many tokens to input.

[4]Our definition of complexity class solvable by circuits with oracle is slightly different from that in literature (Wilson, 1985), where the size of the oracle circuits refers to the number of wires, whereas ours refers to the number of gates.

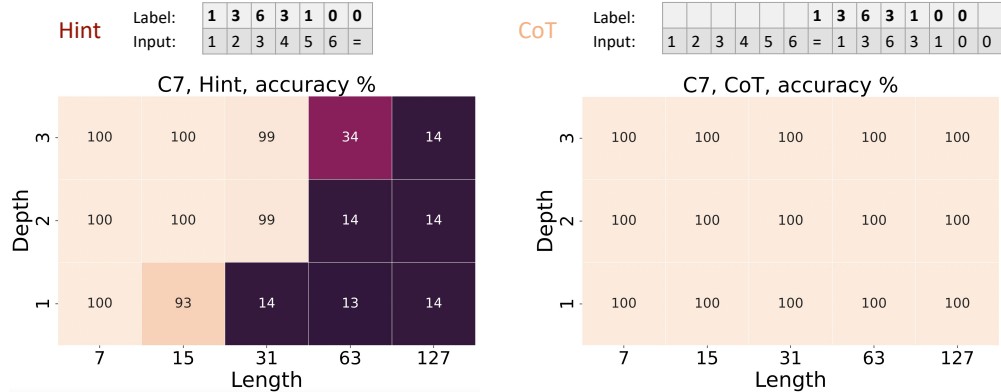

**Figure 2:** Modular Addition($C_7$). The label is the sum of the inputs modulo a positive integer, which is 7 in this case. The chain of thoughts and hints are the partial modular sum. Low-depth transformers with hint can solve this task well for a reasonable input sequence length, but with cot the performance is much better, especially with a long input sequence, as predicted by our Theorem 3.3. See experiments for $C_2$ in Figure 5.

**Definition 3.5** (SIZE$^{\mathcal{L}}$). For any decision problem $\mathcal{L}$ and $T(n) \subseteq O(\mathrm{poly}(n))$, we define SIZE$^{\mathcal{L}}(T(n))$ as the set of decision problems $\mathcal{L}'$ such that there exists $p(n) \in \mathrm{poly}(n)$ and circuits $\{C_n\}_{n=1}^{\infty}$ where $C_n$ contains at most $O(T(n))$ AND, OR, NOT, and $\mathcal{L}_{p(n)}$ gates. For a complexity class $\mathcal{C}$, we define SIZE$^{\mathcal{C}}(T(n)) \triangleq \cup_{\mathcal{L} \in \mathcal{C}} \mathrm{SIZE}^{\mathcal{L}}(T(n))$.

**Theorem 3.7.** For any $T(n) \in \mathrm{poly}(n)$, it holds that $\mathsf{SIZE}^{\mathsf{TC}^0}[1 + T(n)] = \mathsf{CoT}[T(n), \mathrm{poly}(n), \log n]$. Specifically, for $T(n) = 0$, we have $\mathsf{TC}^0 = \mathsf{SIZE}^{\mathsf{TC}^0}[1] = \mathsf{CoT}[0, \mathrm{poly}(n), \log n] = \mathsf{T}[\mathrm{poly}(n), \log n]$.

**Theorem 3.8.** For any $T(n) \in \mathrm{poly}(n)$, it holds that $\mathsf{SIZE}^{\mathsf{AC}^0}[1 + T(n)] = \mathsf{CoT}[T(n), \mathrm{poly}(n), 1]$. Specifically, for $T(n) = 0$, we have $\mathsf{AC}^0 = \mathsf{SIZE}^{\mathsf{AC}^0}[1] = \mathsf{CoT}[0, \mathrm{poly}(n), 1] = \mathsf{T}[\mathrm{poly}(n), 1]$.

Theorem 3.8 shows that for $T(n) = \mathrm{poly}(n)$ steps of CoT, using $\mathrm{poly}(n)$ embedding size does not improve expressiveness over using $\log(n)$ embedding size (Theorem 3.3), because $\mathsf{SIZE}^{\mathsf{TC}^0}[\mathrm{poly}(n)] = \mathsf{SIZE}^{\mathsf{AC}^0}[\mathrm{poly}(n)] = \mathsf{SIZE}[\mathrm{poly}(n)]$. However, Theorem 3.9 shows that for any specific polynomial $T(n) = n^k$ steps of CoT, increasing embedding width from $O(\log(n))$ to $\mathrm{poly}(n)$ make transformers strictly more powerful.

**Theorem 3.9.** For any $s(n) = O(\log n)$, $\mathsf{T}[\log n, s(n)] \subsetneq \mathsf{T}[\mathrm{poly}(n), s(n)]$ and for all $k \in \mathbb{N}$, $\mathsf{CoT}[n^k, \log n, s(n)] \subsetneq \mathsf{CoT}[n^k, \mathrm{poly}(n), s(n)]$.

# 4 CoT EMPIRICALLY IMPROVES EXPRESSIVENESS OF LOW-DEPTH TRANSFORMERS ON INHERENTLY SERIAL PROBLEMS

This section is an empirical study of the expressiveness of decoder-only transformers with CoT on four different arithmetic problems: modular addition, permutation composition ($S_5$), iterated squaring, and circuit value problem. The first problem is parallelizable and can be solved by constant-depth transformers with log-precision while the latter three are inherently serial under some standard hardness assumptions in computational complexity or cryptography. As a prediction of our theory, we expect to see a huge improvement in accuracy when CoT is turned on.

**General Setup.** To examine the expressiveness of decode-only transformers with and without CoT on these four types of problems, we train the transformer using Adam (Kingma & Ba, 2014) from random initialization in the online supervised setting for each problem and each different sequence length $n$. At each step, we sample a batch of training data from a distribution $p_n(x)$ where $x = (x_1, \ldots, x_n)$ is training data and $y = f^*(x_1, \ldots, x_n)$ is the label. We always set $x_n$ to be '='. We consider three different settings, base, cot, and hint:

- base: The optimization objective is simply $\ell_{\mathsf{base}}(\theta) \triangleq \mathbb{E}_{x \sim p} \, \ell_{\mathsf{ce}}(p_\theta(x), f^*(x))$.
- cot: We manually design a chain of thought for each instance $x$, which is a string in $\mathcal{V}$ and we denote by $c(x)$. We ensure the last token of $c(x)$ is always equal to the answer, $f^*(x)$. With $\tilde{x} \triangleq (x, c(x))$, the concatenation of $x$ and $c(x)$, and $m$ be the length of $c(x)$, the optimization objective is $\ell_{\mathsf{cot}}(\theta) \triangleq \frac{1}{m} \mathbb{E}_{x \sim p} \sum_{i=n}^{n+m-1} -\ln p_\theta(\tilde{x}_{i+1} \mid \tilde{x}_1, \ldots, \tilde{x}_i))$.
- hint: Even if the transformer has better performance in cot setting than base setting, one may argue that besides the difference expressiveness, cot setting also has a statisti-

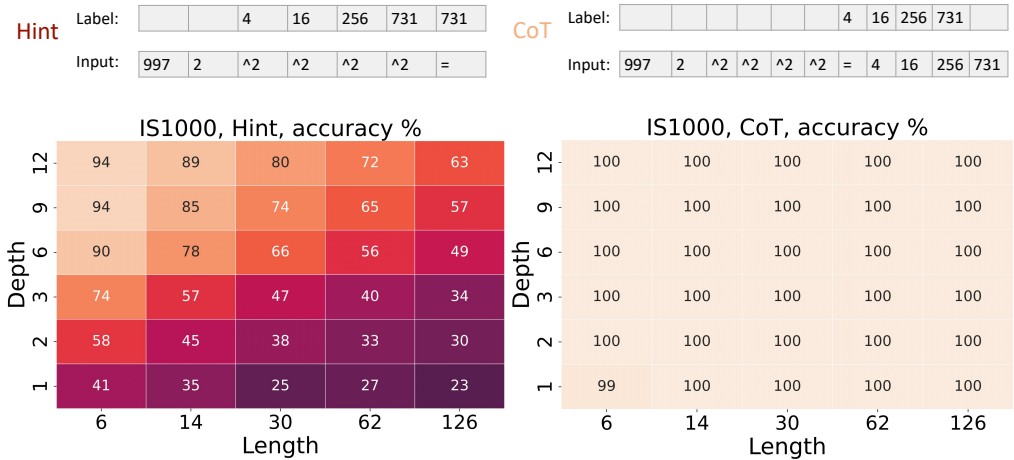

**Figure 3:** Iterated Squaring(IS). The vocabulary $\mathcal{V} \triangleq \{0, 1, \ldots, T-1, =, \hat{}\ 2\}$ with $T = 1000$. We randomly generate input of format $(r, p, \hat{}\ 2, \ldots, \hat{}\ 2, =)$ with $1 \leq r, p \leq T-1$, $p$ being a prime and random number of $\hat{}\ 2$ tokens (at most $m$). The label is $f_{r,p}(n) \equiv (r^{2^n}) \mod p$. CoT and hints are $(f_{r,p}(i))_{i=1}^n$. Though our construction does not exactly satisfy the technical conditions of the hardness assumption, this problem is difficult for transformers without CoT to learn, but can be perfectly expressed with CoT even if depth is only 1.

cal advantage over base, as cot provides more labels and thus more information about the groundtruth $f^*$ to the model. This motivates us to design the following loss which provides the chain of thought $c(x)$ as the labels. Here for simplicity, we assume the length of $c(x)$ is equal to $n$. [5] Formally we define $\ell_{\mathsf{hint}}(\theta) \triangleq \frac{1}{n} \mathbb{E}_{x \sim p} \sum_{i=1}^{n} -\ln p_\theta(c_i(x) \mid x_1, \ldots, x_i))$.

**Performance Evaluation.** Since we train transformers using fresh sampled synthetic data each step, the training accuracy/loss is just the same as validation accuracy/loss. For base and hint setting, we evaluate the accuracy of the final answer directly. For cot setting, directly evaluating the final answer is too easy because it only measures the ability of the transformer to correctly compute the last step since CoT is given as inputs. Ideally, we should measure the answer output by transformers after auto-regressively generating $|c(x)|$ tokens. But for computational efficiency, we measure the probability that transformers can predict all tokens in the given CoT correctly. Note this probability is a lower bound of the ideal metric because there is a small possibility that transformers can answer correctly with a wrong CoT. Nevertheless, even with this slightly more difficult evaluation metric, transformers in cot setting still optimize much faster than without CoT.

Due to space limit, we defer the details of the training and each setting to Appendix A. Our experimental results are presented in Figures 1 to 4.

**Our Findings:** Unsurprisingly, the accuracy in hint setting is always higher than base setting. Due to the space limit, we postpone all results for base settings into Appendix A. For the problems hard for parallel computation, *i.e.*, permutation composition, iterated squaring, and circuit value problem, we find that cot is always better than hintand base, and the improvement is huge especially when depth is small. Our experiments suggest that turning on CoT drastically improves the expressiveness of low-depth transformers on problems which are hard to be parallel computed, *i.e.*, those inherently serial problems.

## 5 RELATED WORKS

Despite the numerous empirical achievements, unanswered questions concerning the inner workings of neural networks capable of algorithmic reasoning. The ability of self-attention to create low-complexity circuits has been recognized (Edelman et al., 2022; Hahn, 2020; Merrill et al., 2021), as well as its capacity to form declarative programs (Weiss et al., 2021), and Turing machines (Dehghani et al., 2018; Giannou et al., 2023; Pérez et al., 2021). Moreover, it has been demonstrated that interpretable symbolic computations can be drawn from trained models (Clark et al., 2019; Tenney et al., 2019; Vig, 2019; Wang et al., 2022b).

Liu et al. (2022a) is a closely related work to ours, which studies the expressiveness of low-depth transformers for semi-automata. Their setting corresponds to using only 1 step of CoT and our contribution is to show that allowing more steps of CoT enables the transformers to solve more

---

[5]Note such alignment is in general impossible because the CoT $c(x)$ can be even longer than $x$ itself. However, in our four settings, there exist meaningful ways to set CoT $c(x)$ as hints for earlier tokens in $x$.

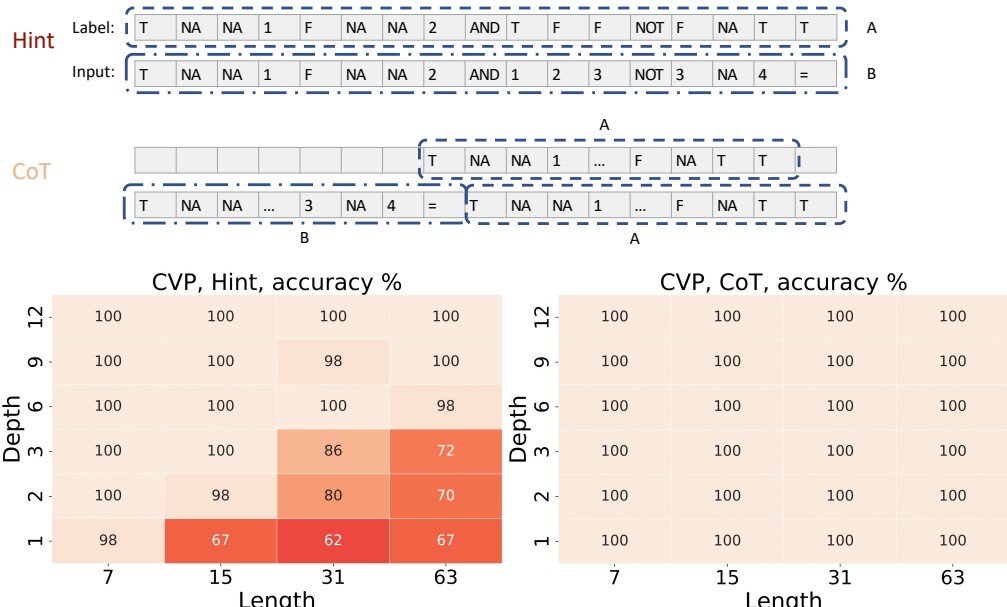

**Figure 4:** Circuit Value Problem(CVP). Given a randomly generated circuit with $m$ gates (sorted by topological order), the vocabulary $\mathcal{V} = [m] \cup \{\text{TRUE}, \text{FALSE}, \text{AND}, \text{OR}, \text{NOT}, \text{NA}, =\}$. Each gate is represented by four consecutive tokens, which are gate type, two input gate ids, and the current gate id. The output is the value of the last gate $m$. CoT and hints also contain 4 tokens for each gate, which are gate type, two input gate values, and the current gate value.

difficult problems than semi-automata, especially those inherently serial problems, like the circuit value problem, which is P-complete.

**Constant precision versus logarithmic precision:** We note that most previous literature on the expressiveness of transformers focuses on the setting of logarithmic precision, including (Merrill & Sabharwal, 2023b; Merrill et al., 2022; 2021; Liu et al., 2022a), *etc*. One main reason as argued by Merrill & Sabharwal (2023a) is that log precision allows the transformer to use uniform attention over the rest tokens. However, recent advancements in LLMs showed that uniform attention might not be necessary towards good performance, at least for natural language tasks. For example, one of the most successful open-sourced LLM, LLAMA 2 (Touvron et al., 2023) takes the input of a sequence of 4096 tokens and uses BF16 precision, which has 1 sign bit, 8 exponent bits and 7 mantissa bits (plus one extra leading bit). As a consequence, for example, BF16 cannot express any floating-point number between $2^8 = 256$ and $2^8 + 2 = 258$, which makes LLAMA 2 impossible to compute uniform attention over 257 elements.

A concurrent work Feng et al. (2023) also studies the benefit of CoT via the perspective of expressiveness, where they show with CoT, transformers can solve some specific P-complete problem. Our result is stronger in the sense that we give a simple and clean construction for each problem in P/poly. We also note the slight difference in the settings, while we mainly focus on constant-precision transformers with $O(\log n)$ embedding size, they focus on $O(\log(n))$ precision transformers with bounded embedding size.

## 6 CONCLUSION

We study the capability of CoT for decoder-only transformers through the lens of expressiveness. We adopt the language of circuit complexity and define a new complexity class $\mathsf{CoT}[T(n), d(n), s(n), e(n)]$ which corresponds to a problem class solvable by constant-depth, constant-precision decoder-only transformers with $O(T(n))$ steps of CoT, $O(d(n))$ embedding size and floating-point numbers with $O(e(n))$ bits of exponents and $O(s(n))$ bits of significand. Our theory suggests that increasing the length of CoT can drastically make transformers more expressive. We also empirically verify our theory in four arithmetic problems. We find that for those three inherently serial problems, transformers can only express the groundtruth function by using CoT.

## ACKNOWLEDGEMENT

We thank Wei Zhan and Lijie Chen for providing references on circuit complexity and various inspiring discussions. We thank Cyril Zhang and Bingbin Liu for helpful discussion on Khron-Rhodes Decomposition Theorem. We thank Kaifeng Lyu for his helpful feedbacks.

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

CONTENTS

# A    ADDITIONAL EXPERIMENTAL RESULTS

In this section present the experimental results for base setting which is omitted in the main paper and the details of training and each task. We use the nanogpt[6] codebase for language modeling.

**Training Details.** For all settings we use Adam with $10^{-5}$ learning rate, $0$ weight decay, $\beta_1 = 0.9$, $\beta_2 = 0.95$, and gradient clipping with threshold equal to $1.0$. The total training budget is $10^6$ steps and we use a linear warmup in the first $2000$ steps starting from $10^{-6}$. For each step, we use a fresh sampled batch of size $64$ from population distribution. We turn off dropout and use float16. We vary the depth of the transformer for different settings while the embedding size and the number of attention heads are fixed to be $512$ and $8$ respectively.

Below we present the setting and the experimental results of each problem respectively.

**Modular Addition** ($C_p$). Given any positive integer $p$, the vocabulary of modular addition problem is $\{0, 1, \ldots, p - 1, =\}$. We generate $x = (x_1, \ldots, x_n)$ in the following way: for each $i \in [n - 1]$, we independently sample $x_i$ from $\{0, 1, \ldots, p - 1\}$ and set $x_n =$ '$=$'. The label is $f^*(x) = \sum_{i=1}^{n-1} x_i \mod p$ and CoT $c(x)$ is $(\sum_{i=1}^{k} x_i \mod p)_{k=1}^{n-1}$. Unsurprisingly, this task is an easy task for transformers because attention can easily express the average function across different positions, and so is the sum function. Then the feedforward layers can compute the modulus of the sum and $m$. We note that the high training accuracy here is not contradictory with our Theorem 3.1, because our sequence length is not long enough and float16 is like log-precision. This intuitive argument is elegantly extended to all solvable groups by leveraging Khron-Rhodes decomposition theorem by Liu et al. (2022a).

**Permutation Composition** ($S_p$). Given any $p \in \mathbb{N}^+$, the vocabulary of permutation composition problem is $\{1, \ldots, p, (, ), =\}$. We pick $n = (p + 2)m + 1$ and generate $x = (x_1, \ldots, x_n)$ in the following way: for each $i \in [m]$, we set $x_{(p+2)(i-1)+1}$ as '(', $x_{(p+2)i}$ as ')' and independently sample a random permutation over $[p]$, $\sigma_i = (x_{(p+2)(i-1)+2}, \ldots, x_{(p+2)(i-1)+p+1})$. We set $x_n$ to be '$=$'. Different from other settings which only have the label at one position, we have $p$ labels for this setting, which is the composition of $\sigma_1 \circ \ldots \circ \sigma_n$. The CoT $c(x)$ is the partial composition from $\sigma_1$ to $\sigma_n$.

As mentioned in Section 3, unless $\mathsf{TC}^0 = \mathsf{NC}^1$, composition of $S_p$ cannot be computed by $\mathsf{TC}^0$ for any $p \geq 5$, since composition of $S_p$ implies the wording problem of $S_p$, which is $\mathsf{NC}^1$-complete under $\mathsf{AC}^0$ reductions. Since all constant-depth poly-embedding-size transformers can be simulated by $\mathsf{TC}^0$ circuits (Theorem 3.2), shallow transformers are not able to solve the composition problem of $S_p$ for $p \geq 5$. Our experimental results in Figure 1 matches this theoretic prediction very well.

**Iterated Squaring (IS).** Iterated squaring refers to the following problem: given integers $r, n, p$, we define the iterated squaring function $f_{r,p}(n) \triangleq r^{2^n} \mod p$. It is often used as hardness assumptions in cryptography (Rivest et al., 1996; Lombardi & Vaikuntanathan, 2020) that iterated squaring cannot be solved in $n - o(n)$ time even with polynomial parallel processors under certain technical conditions (*e.g.*, $p$ is the product of two primes of a certain magnitude and there is some requirement on the order of $r$ as an element of the multiplicative group of integers modulo $p$). In other words, people conjecture there is no faster parallel algorithm than doing squaring for $n$ times.

**Circuit Value Problem (CVP).** Circuit value problem is the computational problem of computing the output of a given Boolean circuit on a given input. It is complete for P under $\mathsf{AC}^0$-reductions. This means if one can solve CVP with constant-depth transformers (or any $\mathsf{TC}^0$ circuits), then any problem in P becomes solvable by $\mathsf{TC}^0$, which is widely believed to be impossible.

---

[6] https://github.com/karpathy/nanoGPT

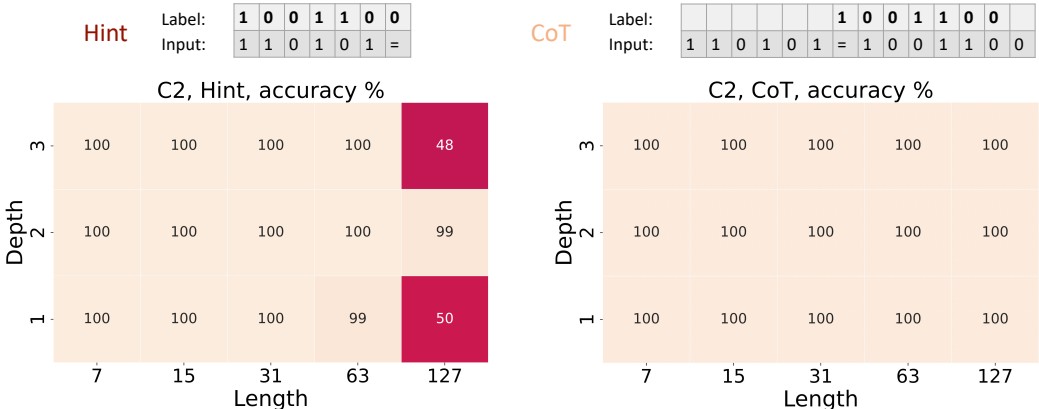

**Figure 5:** Results of Modular Addition $C_2$.

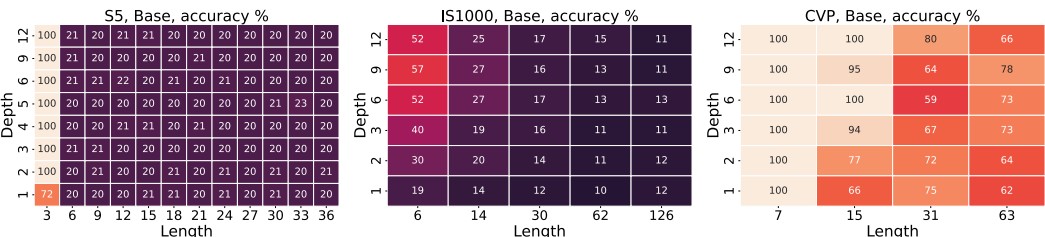

**Figure 6:** Results of base on Permutation Composition, Iterated Squaring, and Circuit Value Problem.

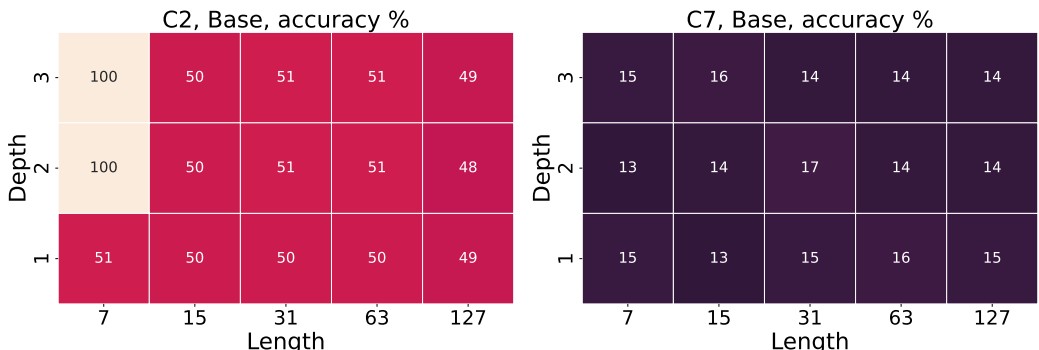

**Figure 7:** Results of Modular Addition base on $C_2$ and $C_7$.

We can observe that the accuracy of base setting is also lower than that of hint setting.

## B ADDITIONAL NOTATIONS AND PRELIMINARIES

We use $\mathbb{N}$ and $\mathbb{R}$ to denote the set of natural numbers and real numbers respectively. For any $n \in \mathbb{N}^+$, we define $[n] \triangleq \{1, 2, \ldots, n\}$. We define $\text{relu}(x) \triangleq \max(x, 0)$. For vector $x$, we use $x_{a:b}$ to denote the vector containing coordinates of $x$ from position $a$ to position $b$. For matrix $M$, we define $M_{a_1:b_1, a_2:b_2}$ to denote the submatrix by selecting rows from $a_1$ to $b_1$, columns from $a_2$ to $b_2$. We also use $a_1$ : to denote the subset of indices from $a_1$ to the end, $: b_1$ to denote the subset of indices from the beginning (1) to $b_1$ and : to denote all indices. Given two non-negative functions $f, g$, we say $f(n) = O(g(n))$ (resp. $f(n) = \Omega(g(n))$) iff there exists $C > 0$, such that for all $n \geq 0$,

$f(n) \leq Cg(n)$ (resp. $f(n) \geq Cg(n)$). We use $\mathsf{poly}(n) \triangleq \{T : \mathbb{N} \to \mathbb{N} \mid \exists k > 0, T(n) = O(n^k)\}$ to denote the set of functions with at most polynomial growth rate.

Given a string $x$ or a set $x$, we use $|x|$ to denote the size of $x$. We use $\phi(x) = \sum_{i=1}^{|x|} 2^{|x|-i} x_i$ to denote the value of binary number represented by binary string $x$. We use $\mathsf{bin}_k(x)$ to denote the usual binary encoding of natural number $x$ using $k$ binary bits in the sense that $\phi(\mathsf{bin}_k(x)) = x$ and $\mathsf{sbin}_k(x)$ to denote the signed binary encoding, which is $2\mathsf{bin}_k(x) - (1, \dots, 1)$. For any $n \in \mathbb{N}^+$, we define $\mathrm{softmax} : \mathbb{R}^n \to \mathbb{R}^n$ as $(\mathrm{softmax}(x))_i = \exp(x_i) / \sum_{i=1}^{n} \exp(x_i)$ for any $x \in \mathbb{R}^n$ and $i \in [n]$. We use $\odot$ to denote the element-wise product of two vectors. We use $a\|b$ or $(a, b)$ to denote the concatenation of two vectors $a$ and $b$.

## B.1 CIRCUIT COMPLEXITY

**Problem.** In this paper we consider the following notion of problems: given a sequence of input tokens, output a token as the answer. Mathematically, given a vocabulary $\mathcal{V}$, we call a mapping $\mathcal{L} : \cup_{k \in \mathbb{N}^+} \mathcal{V}^k \to \mathcal{V}$ a *problem*. If the correct answer is always 0 or 1, we call $\mathcal{L}$ a decision problem. In circuit complexity, such $\mathcal{L}$ is also called a *language*.

Though the standard definition of circuit complexity only deals with binary strings, given any finite vocabulary $\mathcal{V}$, we can always replace each token in $\mathcal{V}$ by its binary representation, and the length of the input only blows up by a constant factor. Therefore we can extend existing complexity classes listed to arbitrary finite vocabulary naturally.

**P.** The class P contains all problems solvable by a deterministic Turing machine in polynomial time.

**Boolean Circuit.** A Boolean circuit over $n$ variables is a directed acyclic graph where nodes are AND, OR, or NOT gates. The gates with in-degree 0 are the inputs, which are assigned one of the $n$ boolean variables. Given the inputs, the circuit computes the value of each non-input gate based on the value of the incoming gates and outputs a number at the output gate.

**SIZE$[T(n)]$.** Given any function $T$, $\mathsf{SIZE}[T(n)]$ denotes the class of problems that can be solved by boolean circuits with $O(T(n))$ gates when the input length is $n$. Formally, a problem $\mathcal{L}$ is in $\mathsf{SIZE}[T(n)]$ if and only if there exists a sequence of circuits $\{C_n\}$ such that each circuit $C_n$ has $n$ inputs and 1 output, the size of each circuit $C_n$ is at most $O(T(n))$, and for all strings $x$, $x$ is in $L$ if and only if $C_{|x|}(x) = 1$.

**P/poly.** We define the class P/poly as the set of problems that can be solved by a family of polynomial-size circuits, that is, $\mathsf{P/poly} \triangleq \cup_{k \in \mathbb{N}^+} \mathsf{SIZE}[n^k]$. Since any Turing Machine with time bound $T(n)$ can be simulated by a circuit of size $T(n) \log T(n)$ (Pippenger & Fischer, 1979), we know that $\mathsf{P} \subseteq \mathsf{P/poly}$.

**NC, AC, and TC.** The class NC contains all problems that can be solved in a small **parallel** runtime—polylogarithmic in input length—and with a polynomial number of processors. Formally, for a positive integer $k$, a problem $\mathcal{L}$ is in $\mathsf{NC}^k$ if and only if there exists a polynomial $p(n)$ and a family of circuits $\{C_n\}$ such that each circuit $C_n$ has $n$ inputs and 1 output, the fan-in of the gates is at most 2, the size of each circuit $C_n$ is at most $p(n)$, the depth of each circuit $C_n$ is $O((\log n)^k)$, and for all strings $x$, $x$ is in if and only if $C_{|x|}(x) = 1$. Finally we define $\mathsf{NC} = \cup_{k \in \mathbb{N}} \mathsf{NC}^k$. The class $\mathsf{AC}^k$ is defined almost the same as $\mathsf{NC}^k$ for each $k \in \mathbb{N}^+$, except the AND and OR gates in $\mathsf{AC}^k$ allow unbounded fan-in. The class $\mathsf{TC}^k$ allows a more powerful type of gate, MAJORITY, compared to $\mathsf{AC}^k$. MAJORITY gate can have unbounded fan-in and is defined as $\mathsf{MAJORITY}(x_1, \dots, x_n) = \lfloor \frac{1}{2} + \frac{(\sum_{i=1}^n x_i) - 1/2}{n} \rfloor$.

It holds that $\mathsf{NC}^i \subseteq \mathsf{AC}^i \subseteq \mathsf{TC}^i \subseteq \mathsf{NC}^{i+1}$ for all natural number $i$. Therefore $\mathsf{NC} = \mathsf{AC} = \mathsf{TC}$, which all stands for the problem class that can be solved in polylogarithmic time with polynomial parallel processors.

## B.2 TRANSFORMER LAYERS

Throughout this paper, we use $d$ to denote the embedding size of a transformer.

**Self-Attention Mechanism:** Given attention parameter $\theta_{\mathsf{ATTN}} = (W_Q, W_K, W_V, W_O) \in \mathbb{R}^{d \times d} \times \mathbb{R}^{d \times d} \times \mathbb{R}^{d \times d} \times \mathbb{R}^{d \times d}$, we define the Attention layer with mask for decoder-only transformer in Algorithm 3. Note allowing multi-head attention will not change the class of problems solvable by constant layer decoder-only transformers as we can simulate 1 multi-head attention layer with any constantly many heads with multiple single-head attention layers. Thus for simplicity of presentation, we do not include multi-head attention in the definition below.

---

**Algorithm 2** Causal Self-Attention, $\mathsf{ATTN}$

---

**Input:** Parameter $\theta_{\mathsf{ATTN}} = (W_Q, W_K, W_V, W_O)$, Input embedding $h = (h_1, \ldots, h_n) \in \mathbb{R}^{nd}$.
**Output:** Output embedding $h' = (h'_1, \ldots, h'_n) \triangleq \mathsf{ATTN}_{\theta_{\mathsf{ATTN}}}(h_1, \ldots, h_n)$.
  1: $q_i \triangleq W_Q h_i, k_i \triangleq W_K h_i, v_i \triangleq W_V h_i, \forall i \in [n]$
  2: $s_i \triangleq \mathrm{softmax}(\langle q_i, k_1 \rangle, \ldots, \langle q_i, k_i \rangle) \| (0, \ldots, 0)$.
  3: $h'_i \triangleq W_O \sum_{j=1}^{n} (s_i)_j v_j$.

---

**Feed-Forward Network:** Given the parameter of fully-connected feedforward network layer $\theta_{\mathsf{FF}} = (W_1, b_1, W_2, b_2) \in \mathbb{R}^{d \times d} \times \mathbb{R}^d \times \mathbb{R}^{d \times d} \times \mathbb{R}^d$, we define the fully-connected feedforward layer $\mathsf{FF}_{\theta_{\mathsf{FF}}} : \mathbb{R}^d \to \mathbb{R}^d$ as $\mathsf{FF}_{\theta_{\mathsf{FF}}}(h) \triangleq W_2 \mathrm{relu}(W_1 h + b_1) + b_2$.

**Token Embedding:** Given the parameter of token embedding layer $\theta_{\mathsf{TE}} \in \mathbb{R}^{d \times |\mathcal{V}|}$, we define the token embedding layer by viewing $\theta_{\mathsf{TE}}$ as a mapping from $\mathcal{V}$ to $\mathbb{R}^d$, that is, for all $x \in \mathcal{V}$, the token embedding is $\theta_{\mathsf{TE}}(x)$.

**Position Encoding:** Given the parameter of position encoding layer $\theta_{\mathsf{PE}} \in \mathbb{R}^{d \times n_{\max}}$, we define the token embedding layer by viewing $\theta_{\mathsf{PE}}$ as a mapping from $[n_{\max}]$ to $\mathbb{R}^d$ that is, for all $n \in [n_{\max}]$, the position embedding is as $\theta_{\mathsf{PE}}(n)$.

**Output Layer:** Given the parameter of output layer $\theta_{\mathsf{OUTPUT}} \in \mathbb{R}^{|\mathcal{V}| \times d}$, we define the output layer $\mathsf{OUTPUT}_{\theta_{\mathsf{OUTPUT}}} : \mathbb{R}^d \to \mathcal{V}$ as $\mathsf{OUTPUT}_{\theta_{\mathsf{OUTPUT}}}(h) \triangleq \mathrm{softmax}(\theta_{\mathsf{OUTPUT}} h)$ for all $h \in \mathbb{R}^d$.

## B.3 Floating-point Numbers

Below we give a formal definition the *floating-point number* and *rounding* operation. Recall we use $\phi(a) = \sum_{i=1}^{k} 2^{k-i} a_i$ to denote the value of binary number represented by $a \in \{0,1\}^k$ for any $k \in \mathbb{N}^+$.

**Definition B.1** (Floating-point Representation)**.** Let $e$ be the number of bits for exponents and $s$ be the number of bits for significand. A $(e + 2s + 1)$-bit binary string $a = (a_1, a_2, \ldots a_{e+2s+1}) \in \{0,1\}^{e+2s+1}$ is a *floating-point* binary representation of number $\phi_{e,s}(a) \triangleq \mathrm{sign}(a) \cdot 2^{\mathrm{exponent}(a)} \cdot \mathrm{significand}(a)$ with $e$-bit exponent and $2s$-precision, where the sign is $\mathrm{sign}(a) \triangleq 2a_1 - 1$, the significand is $\mathrm{significand}(a) \triangleq 2^{-s} \phi(a_{2:2s+1})$, and the exponent is $\mathrm{exponent}(a) \triangleq \phi(a_{2s+2:2s+e+1}) - 2^{\max(0,e-1)}$. We further use $\mathbb{F}_{e,s}$ to denote all the floating numbers representable using $e$-bit exponent and $2s$-bit precision (significand), that is, $\mathbb{F}_{e,s} \triangleq \{ S \cdot 2^{-s+E} \mid -2^{2s} + 1 \leq S \leq 2^{2s} - 1, -2^{\max(0,e-1)} \leq E \leq 2^e - 1 - 2^{\max(0,e-1)}, E, S \in \mathbb{N} \}$. We define $B_{e,s} \triangleq \max \mathbb{F}_{e,s}$.

We also use $\psi_{e,s} : \mathbb{F}_{e,s} \to \{0,1\}^{e+2s+1}$ to denote the inverse of $\phi_{e,s}$. We note that when the number of exponent bits is larger than 0, there are multiple ways to represent a number in $\mathbb{F}_{e,s}$ by a binary string and we assign $\psi_{e,s}(x)$ as the string $a \in \{0,1\}^{e+2s+1}$ with the smallest $|\mathrm{exponent}(a)|$, which is unique for all non-zero numbers. For 0 we additionally set $\mathrm{sign}(\psi_{e,s}(0)) = 1$.

**Definition B.2** (Correct Rounding)**.** For any $x \in \mathbb{R}$ and any closed subset of $\mathbb{R}$ containing 0, $\mathbb{F}$, we define *correct rounding* $\mathrm{round}(x, \mathbb{F})$ as the closest number to $x$ in $\mathbb{F}$. We break the tie by picking the one with a smaller absolute value.

In particular, we denote the rounding operation with $e$-bit exponent, $2s$-bit precision by $\mathrm{round}_{e,s}(\cdot) \triangleq \mathrm{round}(\cdot, \mathbb{F}_{e,s})$, which is also denoted by $[\cdot]_{e,s}$ for convenience. We extend the definition of round and $\mathrm{round}_{e,s}$ to vector inputs by rounding coordinate-wisely.

Our notion of floating-point number simplifies the IEEE 754 Standard for Floating-point Arithmetic (IEEE, 2008) by removing $\infty$ and $-\infty$. When overflow happens, we always round the out-

put to the (negative) largest representable number in $\mathbb{F}_{e,s}$. For unary functions like $\exp(\cdot)$ and binary functions including addition, subtraction, multiplication, and division, we simply define their rounded version by rounding their outputs. Whenever division by $0$ happens, we treat it as the model outputs the wrong result.

## C    DETAILS ON FINITE-PRECISION LAYERS

In this section, we give the definition of the finite-precision version of different transformer layers. Recall that given $s \in \mathbb{N}^+$, the numbers representable using $2s$-bit significand and $e$-bit exponent is $\mathbb{F}_{e,s} \triangleq \{S \cdot 2^{-s+E} \mid -2^{2s}+1 \leq S \leq 2^{2s}-1, -2^{\max(0,e-1)} \leq E \leq 2^e-1-2^{\max(0,e-1)}, E, S \in \mathbb{N}\}$.

**Self-Attention Mechanism:** Given attention parameter $\theta_{\mathsf{ATTN}} = (W_Q, W_K, W_V, W_O) \in \mathbb{F}_{e,s}^{d \times d} \times \mathbb{F}_{e,s}^{d \times d} \times \mathbb{F}_{e,s}^{d \times d} \times \mathbb{F}_{e,s}^{d \times d}$, we define the self-attention layer with causal mask for decoder-only transformer in Algorithm 3.

---

**Algorithm 3** Finite-Precision Causal Self-Attention, ATTN

---

**Input:** Integer $s \in \mathbb{N}^+$, $e \in \mathbb{N}$, Parameter $\theta_{\mathsf{ATTN}} = (W_Q, W_K, W_V, W_O)$, Input embedding $h = (h_1, \ldots, h_n) \in \mathbb{F}_{e,s}^{nd}$.

**Output:** Output embedding $h' = (h'_1, \ldots, h'_n) \triangleq \mathsf{ATTN}_{\theta_{\mathsf{ATTN}}}(h_1, \ldots, h_n)$.

1: $q_i \triangleq W_Q \times_{e,s} h_i, k_i \triangleq W_K \times_{e,s} h_i, v_i \triangleq W_V \times_{e,s} h_i, \forall i \in [n]$

2: $s_i \triangleq \mathsf{softmax}_{e,s}(\langle q_i, k_1 \rangle_{e,s}, \ldots, \langle q_i, k_i \rangle_{e,s}) \| (0, \ldots, 0). \qquad \triangleright n - i\text{'s } 0; \text{ Mask for Causal Attention};$

3: $h'_i \triangleq W_O \times_{e,s} \mathsf{sum}_{e,s}([v_1, \ldots, v_n] \times_{e,s} s_i).$

---

**Feed-Forward Network:** Given $s \in \mathbb{N}^+$, $e \in \mathbb{N}$, and the parameter of fully-connected feedforward network layer $\theta_{\mathsf{FF}} = (W_1, b_1, W_2, b_2) \in \mathbb{F}_{e,s}^{d \times d} \times \mathbb{F}_{e,s}^d \times \mathbb{F}_{e,s}^{d \times d} \times \mathbb{F}_{e,s}^d$, we define the fully-connected feedforward layer $\mathsf{FF}_{\theta_{\mathsf{FF}}} : \mathbb{F}_{e,s}^d \to \mathbb{F}_{e,s}^d$ as $\mathsf{FF}_{\theta_{\mathsf{FF}}}(h) \triangleq \left[ W_2 \times_{e,s} \mathsf{relu}([W_1 \times_{e,s} h + b_1]_{e,s}) + b_2 \right]_{e,s}$.

**Token Embedding:** Given $s \in \mathbb{N}^+$, $e \in \mathbb{N}$, and the parameter of token embedding layer $\theta_{\mathsf{TE}} \in \mathbb{F}_{e,s}^{d \times |\mathcal{V}|}$, we define the token embedding layer by viewing $\theta_{\mathsf{TE}}$ as a mapping from $\mathcal{V}$ to $\mathbb{R}^d$, that is, for all $x \in \mathcal{V}$, the token embedding is $\theta_{\mathsf{TE}}(x)$.

**Position Encoding:** Given $s \in \mathbb{N}^+$, $e \in \mathbb{N}$, and the parameter of position encoding layer $\theta_{\mathsf{PE}} \in \mathbb{F}_{e,s}^{d \times n_{\max}}$, we define the token embedding layer by viewing $\theta_{\mathsf{PE}}$ as a mapping from $[n_{\max}]$ to $\mathbb{R}^d$ that is, for all $n \in [n_{\max}]$, the position embedding is as $\theta_{\mathsf{PE}}(n)$.

**Output Layer:** Given $s \in \mathbb{N}^+$, $e \in \mathbb{N}$, and the parameter of output layer $\theta_{\mathsf{OUTPUT}} \in \mathbb{F}_{e,s}^{|\mathcal{V}| \times d}$, we define the output layer $\mathsf{OUTPUT}_{\theta_{\mathsf{OUTPUT}}} : \mathbb{F}_{e,s}^d \to \mathcal{V}$ as $\mathsf{OUTPUT}_{\theta_{\mathsf{OUTPUT}}}(h) \triangleq \mathsf{softmax}_{e,s}(\theta_{\mathsf{OUTPUT}} \times_{e,s} h)$ for all $h \in \mathbb{F}_{e,s}^d$.

Finally, we define finite-precision decoder-only transformers below.

## D    PRELIMINARY OF AUTOMATA AND KROHN-RHODES DECOMPOSITION THEOREM

In this section we recap the basic notations and definitions for automata theory and Krohn-Rhodes Decomposition Theorem (Krohn & Rhodes, 1965), following the notation and presentation of Maler (2010).

**Definition D.1** (Automaton). A deterministic automaton is triple $\mathcal{A} = (\Sigma, Q, \delta)$ where $\Sigma$ is a finite set of symbols called the input alphabet, $Q$ is a finite set of states and $\delta : Q \times \Sigma \to Q$ is the transition function.

The transition function can be lifted naturally to input sequences, by letting $\delta(q, w\sigma) \triangleq \delta(\delta(q, w), \sigma)$ for all $w \in \Sigma^*$ recursively.

---

**Algorithm 4** Finite-precision Decoder-only Transformer, $\mathsf{TF}_\theta$ and $p_\theta$

---

**Input:** Integer $s \in \mathbb{N}^+$, $e \in \mathbb{N}$. Transformer parameter $\theta = (\theta_{\mathsf{PE}}, \theta_{\mathsf{TE}}, \theta_{\mathsf{OUTPUT}}, \{\theta_{\mathsf{ATTN}}^{(l)}, \theta_{\mathsf{FF}}^{(l)}\}_{l=0}^{L-1})$
   with $2s$-bit precision and $e$-bit exponent. Input tokens $x = (x_1, \ldots, x_n) \in \mathcal{V}^n$.
**Output:** Output distribution $p_\theta(\cdot \mid x_1, \ldots, x_i)$ for all $i \in [n]$ and output token $\mathsf{TF}_\theta(x)$.
1: $h_i^{(0)} \triangleq [\mathsf{TE}(x_i) + \mathsf{PE}(i)]_{e,s}, \forall i \in [n]$
2: **for** $l = 0, \ldots, L - 1$ **do**
3: $\quad (h_1^{(l+0.5)}, \ldots, h_n^{(l+0.5)}) \triangleq \left[ (h_1^{(l)}, \ldots, h_n^{(l)}) + \mathsf{ATTN}_{\theta_{\mathsf{ATTN}}^{(l)}}(h_1^{(l)}, \ldots, h_n^{(l)}) \right]_{e,s}$
4: $\quad h_i^{(l+1)} \triangleq \left[ h_i^{(l+0.5)} + \mathsf{FF}_{\theta_{\mathsf{FF}}^{(l)}}(h_i^{(l+0.5)}) \right]_{e,s}, \forall i \in [n]$
5: **end for**
6: $p_\theta(\cdot \mid x_1, \ldots, x_i) \triangleq \left[ \mathsf{OUTPUT}_{\theta_{\mathsf{OUTPUT}}}(h_i^{(L)}) \right]_{e,s}, \forall i \in [n]$
7: $\mathsf{TF}_\theta(x) \triangleq \arg\max_x p_\theta(x \mid x_1, \ldots, x_n)$.

---

An automaton can be made an acceptor by choosing an initial state $q_0 \in Q$ and a set of accepting states $F \subseteq Q$. As such it accepts/recognizes a set of sequences, also known as a language, defined as $\mathcal{L}(A, q_0, F) = \{w \in \Sigma^* : \delta(q_0, w) \in F\}$. Kleene's Theorem states that the class of languages recognizable by finite automata coincides with the regular languages.

**Definition D.2** (Automaton Homomorphism). A surjection $\phi : Q \to Q_0$ is an automaton homomorphism from $\mathcal{A} = (\Sigma, Q, \delta)$ to $\mathcal{A}_0 = (\Sigma, Q_0, \delta_0)$ if for every $q \in Q, \sigma \in \Sigma, \phi(\delta(q, \sigma)) = \delta_0(\phi(q), \sigma)$. In such a case we say that $\mathcal{A}_0$ is homomorphic to $\mathcal{A}$ and denote it by $\mathcal{A}_0 \leq_\phi \mathcal{A}$. When $\phi$ is a bijection, $\mathcal{A}$ and $\mathcal{A}_0$ are said to be isomorphic.

The conceptual significance of Automaton Homomorphism is that, if we can simulate any $\mathcal{A}$ and $\mathcal{A}_0 \leq_\phi \mathcal{A}$, we can 'almost' simulate $\mathcal{A}_0$ as well, in the sense of following lemma:

**Lemma D.1.** For any two automata $\mathcal{A} = (\Sigma, Q, \delta), \mathcal{A}_0 = (\Sigma, Q_0, \delta_0)$ satisfying that $\mathcal{A}_0 \leq_\phi \mathcal{A}$ for some function $\phi$, for any $F_0 \subseteq Q$, $q_0 \in Q$, $\phi(q) = q_0$, it holds that $\mathcal{L}(\mathcal{A}_0, q_0, F_0) = \mathcal{L}(\mathcal{A}, q, \phi^{-1}(F_0))$.

*Proof of Lemma D.1.* We claim for any $w \in \Sigma^*$, it holds that $\phi(\delta(q, w)) = \delta(\phi(q), w)$. This claim holds by definition of automaton homomorphism for all $|w| \leq 1$. suppose the claim already holds for all $w$ no longer than $n$ for some $n$, for any $w' = w\sigma$ with $|w| = n$ and $\sigma \in \Sigma$, it holds that $\phi(\delta(q, w')) = \phi(\delta(\delta(q, w), \sigma)) = \delta(\phi(\delta(q, w)), \sigma) = \delta(\delta(\phi(q), w), \sigma) = \delta(\phi(q), w')$. Therefore $\delta_0(q_0, w) \in F_0 \iff \delta_0(\phi(q), w) \in F_0 \iff \phi(\delta(q, w)) \in F_0 \iff \delta(q, w) \in \phi^{-1}(F_0)$. Thus we conclude that $\mathcal{L}(\mathcal{A}_0, q_0, F_0) = \{w \in \Sigma^* \mid \delta(q_0, w) \in F_0\} = \{w \in \Sigma^* \mid \delta(q, w) \in \phi^{-1}(F_0)\} = \mathcal{L}(\mathcal{A}, q, \phi^{-1}(F_0))$. $\qquad\square$

**Definition D.3** (Semigroups, Monoids and Groups). A Semigroup is a pair $(S, \cdot)$ where $S$ is a set and $\cdot$ is a binary associative operation ("multiplication") from $S \times S$ to $S$. A Monoid $(S, \cdot, 1)$ is a semigroup admitting an identity element $1$ such that $s \cdot 1 = 1 \cdots = s$ for every $s \in S$. A group is a monoid such that for every $s \in S$ there exists an element $s^{-1} \in S$ (an inverse) such that $s \cdot s^{-1} = 1$.

**Definition D.4** (Semigroup Homomorphisms). A surjective function $\phi : S \to S_0$ is a semigroup homomorphism from $(S, \cdot)$ to $(S_0, *)$ if for every $s_1, s_2 \in S, \phi(s_1 \cdot s_2) = \phi(s_1) * \phi(s_2)$. In such a case we say that $S_0$ is homomorphic to $S$ and denote it by $S_0 \leq_\phi S$. Two mutually homomorphic semigroups are said to be isomorphic.

**Definition D.5** (Transformation Semigroup). The transformation semigroup of an automata $\mathcal{A} = (\Sigma, Q, \delta)$ is the semigroup generated by $\{\delta(\cdot, \sigma) : Q \to Q \mid \sigma \in \Sigma\}$.

## D.1 THE KROHN-RHODES DECOMPOSITION THEOREM

Below we give the definition of the cascade product of two automata, which is a central concept used in Krohn-Rhodes Decomposition Theorem for automata.

**Definition D.6** (Cascade Product). Let $\mathcal{B}_1 = (\Sigma, Q_1, \delta_1)$ and $\mathcal{B}_2 = (Q_1 \times \Sigma, Q_2, \delta_2)$ be two automata. The cascade product $\mathcal{B}_1 \circ \mathcal{B}_2$ is the automaton $C = (\Sigma, Q_1 \times Q_2, \overline{\delta})$ where

$$\overline{\delta}((q_1, q_2), \sigma) \triangleq (\delta_1(q_1, \sigma), \delta_2(q_2, (q_1, \sigma))).$$

The cascade product of more than two automata is defined as $\mathcal{B}_1 \circ \mathcal{B}_2 \circ \cdots \circ \mathcal{B}_k = (\cdots ((\mathcal{B}_1 \circ \mathcal{B}_2) \circ B_3 \cdots) \circ \mathcal{B}_k$.

**Definition D.7** (Permutation-Reset Automata). A automaton $\mathcal{A} = (\Sigma, Q, \delta)$ is a permutation-reset automaton if for every letter $\sigma \in \Sigma$, $\sigma$ is either a permutation or reset. If the only permutations are identities, we call it a reset automaton.

**Theorem D.2** (Krohn-Rhodes; cf. Maler (2010)). For every automaton A there exists a cascade $\mathcal{C} = \mathcal{B}_1 \circ \mathcal{B}_2 \circ \cdots \circ \mathcal{B}_k$ such that:

1. Each $\mathcal{B}_i$ is a permutation-reset automaton;

2. There is a homomorphism $\phi$ from $\mathcal{C}$ to $\mathcal{A}$;

3. Any permutation group in some $\mathcal{B}_i$ is homomorphic to a subgroup of the transformation semigroup of $\mathcal{A}$.

The pair $(\mathcal{C}, \phi)$ is called a cascaded decomposition of $\mathcal{A}$.

## D.2 COUNTER-FREE AUTOMATA

Next we introduce a key concept used in the proof of Theorem E.1 (and thus Theorem 3.1) – Counter-free Automaton.

**Definition D.8** (Counter-free Automaton, (McNaughton & Papert, 1971)). An automaton is counter-free if no word $w \in \Sigma^*$ induces a permutation other than identity on any subset of $Q$.

A subclass of the regular languages is the class of star-free sets defined as:

**Definition D.9** (Star-Free regular languages). The class of star-free regular languages over $\Sigma$ is the smallest class containing $\Sigma^*$ and the sets of the form $\{\sigma\}$ where $\sigma \in \sigma \cup \{\epsilon\}$, which is closed under finitely many applications of concatenation and Boolean operations including union, intersection, and complementation.

It is well-known that languages recognized by counter-free automata have the following equivalent characterizations.

**Theorem D.3** (McNaughton & Papert (1971)). Suppose $L$ is a regular language not containing the empty string. Then the following are equivalent:

1. $L$ is star-free;

2. $L$ is accepted by a counter-free automata.

3. $L$ is non-counting, *i.e.*, there is an $n \in \mathbb{N}$ so that for all $x$, $y$, and $z$ and all $m \geq n$, $xy^m z \in L \iff xy^{m+1} z \in L$.

Counter-free property of an automaton can also be characterized via its transformation semigroup by Lemma D.4, whose proof is straightforward and skipped.

**Lemma D.4.** An automaton is counter-free if and only if the transformation semigroup of the automaton is group-free, *i.e.*, it has no non-trivial subgroups. A semigroup $(S, \cdot)$ is group-free if and only if it is *aperiodic*, *i.e.*, for all $s \in S$, there exists $k \in \mathbb{N}$, $s^k = s^{k+1}$.

Thus Theorem D.5 holds as a corollary of Theorem D.2.

**Theorem D.5** (Corollary of Theorem D.2). For every counter-free automaton $\mathcal{A}$ there exists a cascade $\mathcal{C} = \mathcal{B}_1 \circ \mathcal{B}_2 \circ \cdots \circ \mathcal{B}_k$ such that each $\mathcal{B}_i$ is a reset automaton and there is a homomorphism $\phi$ from $\mathcal{C}$ to $\mathcal{A}$.

Using Theorem D.5 the following theorem connects the counter-free automata to constant-depth poly-size circuits with unbounded fan-in. The high-level proof idea is that any reset automaton can be simulated using constantly many depth and any counter-free automaton can be decomposed into the cascade product of a finite number of reset automaton.

**Theorem D.6.** [Theorem 2.6, Chandra et al. (1983)] Suppose $\mathcal{A} = (\Sigma, Q, \delta)$ is an counter-free automaton. Then there is a circuit of size $O(n^3)$ with unbounded fan-in and constant depth that simulates $\delta(q, w)$ for any $q \in Q$ and $w \in \Sigma^*$ satisfying $|w| = n$, where $O(\cdot)$ hides constants depending on the automaton.

# E  PROOFS FOR EXPRESSIVENESS UPPER BOUNDS (SECTION 3.3)

The main technical theorems we will prove in this section are Theorems E.1 and E.2. Their proofs can be found in Appendices E.1 and E.2 respectively.

Recall $\psi_{e,s} : \mathbb{F}_{e,s} \to \{0,1\}^{e+2s+1}$ is the binary representation of floating point with $e$-bit exponent and $2s$-bit precision.

**Theorem E.1.** For any fixed $e \in \mathbb{N}, s \in \mathbb{N}^+$, $\mathsf{sum}_{e,s} : (\mathbb{F}_{e,s})^n \to \mathbb{F}_{e,s}$ has $\mathsf{AC}^0$ circuits.

In detail, there is a family of $\mathsf{AC}^0$ circuits $\{C_n\}$ such that for all $x_1, \ldots, x_n \in \mathbb{F}_{e,s}$, it holds that

$$C_n(\psi_{e,s}(x_1) \| \ldots \| \psi_{e,s}(x_n)) = \psi_{e,s}(\mathsf{sum}_{e,s}(x_1, \ldots, x_n)) \tag{2}$$

**Theorem E.2.** For $s(n) = O(\mathsf{poly}(n))$, $\mathsf{sum}_{0,s(n)} : (\mathbb{F}_{0,s(n)})^n \to \mathbb{F}_{0,s(n)}$ has $\mathsf{TC}^0$ circuits.

In detail, there is a family of $\mathsf{TC}^0$ circuits $\{C_n\}$ such that for all $x_1, \ldots, x_n \in \mathbb{F}_{0,s(n)}$, it holds that

$$C_n(\psi_{0,s(n)}(x_1) \| \ldots \| \psi_{0,s(n)}(x_n)) = \psi_{0,s(n)}(\mathsf{sum}_{e,s}(x_1, \ldots, x_n)) \tag{3}$$

With Theorems E.1 and E.2 ready, Theorems 3.1 and 3.2 are standard (e.g., see proof of Theorem 4 in Liu et al. (2022a)) and thus are omitted.

## E.1  PROOFS FOR THEOREM E.1

**Definition E.1** (Total Order). A total order $\leq$ on some set $X$ is a binary relationship satisfying that for all $a, b, c \in X$:

1. $a \leq a$ (reflexive)

2. $a \leq b, b \leq c \implies a \leq c$ (transitive)

3. $a \leq b, b \leq a \implies a = b$ (antisymmetric)

4. $a \leq b$ or $b \leq a$. (total)

**Definition E.2** (Ordered Automaton). We say an automaton $\mathcal{A} = (\Sigma, Q, \delta)$ is *ordered* if and only if there exists a total order $\leq$ on $Q$ and for all $\sigma \in \Sigma$, $\delta(\cdot, \sigma)$ preserves the order, that is,

$$\forall q, q' \in Q, \quad q \geq q' \implies \delta(q, \sigma) \geq \delta(q', \sigma).$$

**Theorem E.3.** All ordered automata are counter-free. Languages recognizable by any ordered automata belong to $\mathsf{AC}^0$.

*Proof of Theorem E.3.* To show an ordered automaton $\mathcal{A} = (\Sigma, Q, \delta)$ is counter-free, it suffices to its transformation semigroup is group-free, or aperiodic. We first recall the definition of aperiodic semigroups Lemma D.4. Let $\pi_w : Q \to Q, \pi_w(q) \triangleq \delta(q, w)$ be the transformation induced by word $w \in \Sigma^*$. Transformation semigroup of $\mathcal{A}$ is aperiodic iff for any $w \in \Sigma^*$, there exists $k \in \mathbb{N}$, such that $(\pi_w)^k = (\pi_w)^{k+1}$.

Now We claim for any $q \in Q$, there is $k \in \mathbb{N}$, such that $(\pi_w)^k(q) = (\pi_w)^{k+1}(q)$. Since $Q$ is finite, this implies that there exists $k \in \mathbb{N}$, such that $(\pi_w)^k = (\pi_w)^{k+1}$ and thus the transformation semigroup of $\mathcal{A}$ is aperiodic. First, note that $\mathcal{A}$ is ordered, we know $\pi_\sigma$ is order-preserving for all $\sigma \in \Sigma$. Let $w = \overline{w_1 \cdots w_n}$ where $|w| = n$, we have $\pi_w = \pi_{w_1} \circ \cdots \circ \pi_{w_n}$ is also order-preserving and thus for all $q \geq q' \in Q$, $\pi_w(q) \geq \pi_w(q')$. Then we proceed by three cases for each $q \in Q$:

1. $\pi_w(q) = q$. In this case, it suffices to take $k = 0$;

2. $\pi_w(q) \geq q$. Since $\pi_w$ is order-preserving, we know for any $k \in \mathbb{N}$, $(\pi_w)^{k+1}(q) \geq (\pi_w)^k(q)$. Since $Q$ is finite, there must exist some $k \in \mathbb{N}$ such that $(\pi_w)^{k+1}(q) = (\pi_w)^k$.

3. $\pi_w(q) \leq q$. Same as the case of $\pi_w(q) \geq q$.

Since $\geq$ is a total order, at least one of the three cases happens. This concludes the proof.

The second claim follows directly from Theorem D.3. $\square$

For any $e, s \in \mathbb{N}$, iterated addition on floating point numbers with $e$-bit exponent and $s$-bit significand $\mathbb{F}_{e,s}$ can be viewed $\mathcal{A}_{e,s} = (\mathbb{F}_{e,s}, \mathbb{F}_{e,s}, \delta_+)$.

**Theorem E.4.** Automaton $\mathcal{A}_{e,s} = (\mathbb{F}_{e,s}, \mathbb{F}_{e,s}, \delta_+)$ is ordered, where $\delta_+(x, y) \triangleq [x + y]_{e,s}$ for any $x, y \in \mathbb{F}_{e,s}$.

*Proof of Theorem E.4.* The total order we use for $\mathbb{F}_{e,s}$ as the state space of automaton $\mathcal{A}$ coincides with the usual order $\leq$ on $\mathbb{R}$. Recall the rounding operation is defined as $[x]_{e,s} \in \arg\min_{x' \in \mathbb{F}_{e,s} |x - x'|}$, which means rounding operation is order preserving, that is, for any $x \geq x' \in \mathbb{F}_{e,s}$, $[x]_{e,s} \geq [x']_{e,s}$. Thus for any $x, x', y \in \mathbb{F}_{e,s}$ with $x \leq x'$, it holds that $\delta_+(x, y) = [x + y]_{e,s} \geq [x' + y]_{e,s} = \delta_+(x', y)$. Thus $\mathcal{A}_{e,s}$ is ordered. □

The following theorem Theorem E.1 is a direct consequence of Theorem E.4.

### E.2 PROOFS FOR THEOREM E.2

We first claim that the following algorithm Algorithm 5 correctly computes $\mathsf{sum}_{0,s(n)}$ over $n$ numbers in $\mathbb{F}_{0,s(n)}$.

**Lemma E.5.** Algorithm 5 outputs $\mathsf{sum}_{0,s(n)}(x_1, \ldots, x_n)$ for all $n \in \mathbb{N}^+$ and $x_1, \ldots, x_n \in \mathbb{F}_{0,s(n)}$.

*Proof of Lemma E.5.* Note that $y_{-2} = 0$, $[y_{-1} + y_0]_{0,s(n)} = \mathsf{sign}(x_1) \cdot B_{s(n)}$, and $[\mathsf{sign}(x_1) \cdot B_{s(n)} + y_1]_{0,s(n)} = x_1$, thus we conclude $\mathsf{sum}_{0,s(n)}(x_1, \ldots, x_n) = \mathsf{sum}_{0,s(n)}(y_{-2}, y_{-1}, y_0, y_1, \ldots, y_n)$. Without loss of generality, we can assume that $x_1 > 0$. Therefore $S_{-2} = 0, S_{-1} = B_{s(n)}$, and $S_0 = 2B_{s(n)}$, which further implies that $L_{-2} \leq 0$ and $U_{-2} \geq 2B_{s(n)}$. This ensures $i^*$ is always well-defined. For convenience we use $H_i$ to denote $\mathsf{sum}_{0,s(n)}(y_{-2}, y_{-1}, y_0, y_1, \ldots, y_{i-1})$ in the rest of this proof.

Now we claim either $S_{i^*} = L_{i^*}$ or $S_{i^*} = U_{i^*}$. By definition of $i^*$, if neither of these two equalities happen, we have that $i^* \leq n - 1$, $U_{i^*} = U_{i^*+1}$, and $L_{i^*} = L_{i^*+1}$, which contradicts with the maximality of $i^*$ since $U_{i^*+1} - L_{i^*+1} = U_{i^*} - L_{i^*} \geq 2B_{s(n)}$. Without loss of generality, we assume $S_{i^*} = L_{i^*}$ and the analysis for the other case is almost the same. Now we claim that for all $i > i^*$, no negative overflow happens at position $i$, that is, $H_i + y_i \geq -B_{s(n)}$.

We will prove this claim for two cases respectively depending on whether there exists some $i^* < j < i$ such that $\mathsf{sum}_{0,s(n)}(y_{-2}, y_{-1}, y_0, y_1, \ldots, y_j) = B_{s(n)}$. The first case is such $j$ does not exist. Then neither positive or negative overflow happens through $i^*$ to $i$, and thus

$$H_{i-1} + y_i = H_{i^*} + (S_i - S_{i^*}) \geq H_{i^*} \geq -B_{s(n)}. \tag{4}$$

If such $j$ exists, we let $j^*$ to be the maximum of such $j$. Then neither positive or negative overflow happens through $j^*$ to $i$. Due to the optimality of $i^*$, we know that for all $i, j \geq i^*$, $|S_i - S_j| < 2B_{s(n)} <$. Thus

$$H_{i-1} + y_i = H_{j^*} + (S_i - S_{j^*}) \geq B_{s(n)} - 2B_{s(n)} \geq -B_{s(n)}. \tag{5}$$

Now we claim $H_{k^*} = B_{s(n)}$. Because there is no negative overflow between $i^*$ and $k^*$, we have that $H_{k^*} - H_{i^*} \geq S_{k^*} - S_{i^*} \geq 2B_{s(n)}$ and the fist inequality is only strict when positive overflow happens at some $i^* \leq j \leq k^*$. If there is no such $j$, then $B_{s(n)} \geq H_{k^*} \geq H_{i^*} + 2B_{s(n)} \geq B_{s(n)}$ and thus $H_{k^*} = B_{s(n)}$. Otherwise such $j$ exists and $j^*$ be the maximum of such $j$. Then $H_{k^*} \geq H_{j^*} + (S_{k^*} - S_{j^*}) \geq B_{s(n)} + (S_{k^*} - S_{j^*}) \geq B_{s(n)}$, where the last inequality is due to the optimality of $k^*$. Thus in both cases we conclude that $H_{k^*} = B_{s(n)}$.

Finally we will show there is neither negative or positive overflow from $k^* + 1$ to $n$ and thus $H_n = H_{k^*} + S_n - S_{k^*}$, which would justify the correctness of the algorithm. We have already shown there is no negative overflow. Suppose there is a positive overflow at some $j > k^*$ in the sense that $H_{j-1} + y_j \geq B_{s(n)}$ and we let $j^*$ be the first positive overflow after $k^*$. By definition of $j^*$, there is neither positive and negative overflow between $k^* + 1$ and $j^*$ and thus $H_{j^*-1} + y_{j^*} = H_{k^*} + (S_{j^*} - S_{k^*}) \leq H_{k^*} = B_{s(n)}$, which is contradictory to the assumption that there is a positive overflow at $j^*$. This concludes the proof. □

---

**Algorithm 5** $\mathsf{AC}^0$ algorithm for iterative addition for poly-precision floating point numbers

---

**Input:** Integer $n \in \mathbb{N}^+$, $s(n) = O(\mathsf{poly}(n))$. Floating numbers $x_1, \ldots, x_n \in \mathbb{F}_{0,s(n)}$.
**Output:** ans $= \mathsf{sum}_{0,s(n)}(x_1, \ldots, x_n) \in \mathbb{F}_{0,s(n)}$.
 1: $y_{-2} \leftarrow 0, y_{-1}, y_0 \leftarrow \mathsf{sign}(x_1) \cdot B_{s(n)}, y_1 \leftarrow x_1 - \mathsf{sign}(x_1) \cdot B_{s(n)}, y_i \leftarrow x_i, \forall i \in \{2, \ldots, n\}$;
 2: $S_i \leftarrow \sum_{j=0}^{i} y_j, \ \forall i \in \{-2, \ldots, n\}$;
 3: $U_i \leftarrow \max_{i \le j \le n} S_j, L_i \leftarrow \min_{i \le j \le n} S_j, \ \forall i \in \{-2, \ldots, n\}$;
 4: $i^* \leftarrow \max\{-2 \le i \le n \mid U_i - L_i \ge 2B_{s(n)}\}$;
 5: **if** $S_{i^*} = U_{i^*}$ **then**
 6: $\quad k^* \leftarrow \max\{i^* \le k \le n \mid S_k = L_{i^*}\}$;
 7: $\quad O \leftarrow -B_{s(n)}$;
 8: **else**
 9: $\quad k^* \leftarrow \max\{i^* \le k \le n \mid S_k = U_{i^*}\}$;
10: $\quad O \leftarrow B_{s(n)}$;
11: **end if**
12: ans $\leftarrow O + S_n - S_{k^*}$.

---

*Proof of Theorem 3.2.* It suffices to show that Algorithm 5 can be implemented by a family of $\mathsf{AC}^0$ circuits since Lemma E.5 guarantees the correctness of Algorithm 5. We can treat all the fixed-point floating numbers in the Algorithm 5 as integers with a suitable rescaling, which is $2^{s(n)}$. Since both sorting and adding $n$ binary integers with polynomial bits have $\mathsf{AC}^0$ circuits, each line in Algorithm 5 can be implemented by an $\mathsf{AC}^0$ circuits (for all indexes $i$ simultaneously if there is any). $\qquad \square$

## F   PROOFS FOR EXPRESSIVENESS LOWER BOUNDS (SECTION 3.4)

We first introduce some notations. Since in the construction of the lower bounds we are only using fixed-point numbers, we will use the shorthand $\mathbb{F}_s \triangleq \mathbb{F}_{0,s} = \{c \cdot k \cdot 2^{-s} \mid c \in \{-1, 1\}, 0 \le k \le 2^{2s} - 1, k \in \mathbb{N}\}$ and rounding operation $[\cdot]_s \triangleq [\cdot]_{0,s}$. We use $1_s$ to denote all-one vectors of length $s$. Similarly we define $\langle \cdot, \cdot \rangle_s$, $\times_s$, and $\mathsf{softmax}_s$. We recall that for any $s \in \mathbb{N}^+$ and integer $0 \le x \le 2^s - 1$, we use $\mathsf{bin}_s(x) \in \{0, 1\}^s$ to denote the usual binary encoding of integer $x$ using $s$ binary bits in the sense that $x = \sum_{i=1}^{s} 2^i (\mathsf{bin}_s(x))_i$ and $\mathsf{sbin}_s(x) \in \{-1, 1\}^s$ to denote the signed binary encoding, which is $2\mathsf{bin}_s(x) - (1, \ldots, 1)$.

Recall $B_s = \max \mathbb{F}_s = 2^s - 2^{-s}$.

**Lemma F.1.** For any $s \in \mathbb{N}^+$, it holds that $[\exp(-B_s)]_s = 0$.

*Proof of Lemma F.1.* By the definition of rounding operation for $2s$-bit precision (Definition B.2), it suffices to show that $\exp(-B_s) \le 2^{-s-1}$, that is, $B_s \ge \ln 2 \cdot (s + 1)$. Note that $2^s \ge s + 1$ for all $s \ge 1$, we have $B_s/(s+1) \ge B_s 2^{-s} = 1 - 2^{-2s} \ge 3/4 \ge \ln 2$. $\qquad \square$

Using the same argument above, we also have Lemma F.2.

**Lemma F.2.** For any $s \in \mathbb{N}^+$, it holds that $[\exp(B_s)]_s = B_s$.

### F.1   PROOF OF THEOREM 3.3

Given two vectors $x, y$ of the same length $s$, we use $x^\frown y$ to denote their interleaving, that is, $(x^\frown y)_{2i-1} = x_i, (x^\frown y)_{2i} = y_i$ for all $i \in [e]$.

**Lemma F.3.** For any $s \in \mathbb{N}^+$, let $q_i = \mathsf{sbin}_s(i)^\frown 1_s$ and $k_i = B_s \cdot (\mathsf{sbin}_s(i)^\frown (-1_s))$ for all $i \in [2^s - 1]$, it holds that $\left[\exp(\langle q_i, k_j \rangle_s)\right]_s = \mathbf{1}[i = j]$ for all $i, j \in [2^s - 1]$.

*Proof of Lemma F.3.* It suffices to prove that $\langle q_i, k_j \rangle_s = -B_s$ if $i \ne j$ and $\langle q_i, k_j \rangle_s = 0$ if $i = j$. The rest is done by Lemma F.1.

Given any $i, j \in [2^s - 1]$, by definition of finite-precision inner product, we know that for any $l \in [2s - 1]$, it holds that $a_l = \left[a_{l-1} + [(q_i)_l (k_j)_l]_s\right]_s$ where $a_0 \triangleq 0$ and $a_l \triangleq \langle (q_i)_{:l}, (k_j)_{:l} \rangle_s$ for $l \in [2s]$.

For all $l \in [e]$, we have that $[(q_i)_{2l}(k_j)_{2l}]_s = -B_s$ and $[(q_i)_{2l-1}(k_j)_{2l-1}]_s = B_s \cdot \mathbb{1}[(\mathsf{sbin}_s(i))_l = (\mathsf{sbin}_s(j))_l]$. If $i = j$, it is straightforward that $a_{2l-1} = -B_s$ and $a_{2l} = 0$ for all $l \in [e]$. If $i \neq j$, then there exists $l \in [s - 1]$ such that $(\mathsf{sbin}(i)))_l \neq (\mathsf{sbin}(j)))_l$. Thus $[(q_i)_{2l-1}(k_j)_{2l-1}]_s = [(q_i)_{2l}(k_j)_{2l}]_s = -B_s$, which implies $a_{2l} = -B_s$ regardless of the value of $a_{2l-2}$. Again use induction we can conclude that $a_{2l'} = -B_s$ for all $l \leq l' \leq e$. □

*Proof of Theorem 3.3.* For any $\mathcal{L} \in \mathsf{SIZE}[T(n)]$, by definition there is a family of boolean circuit $\{C_n\}$ which compute $\mathcal{L}$ for all inputs of length $n$ using $O(T(n))$ many NOT and AND gates. Without loss of generality, let us assume the number of non-input gates in $C_n$ be $T(n)$. We will show that for each $n$, there is a 2-layer decoder-only transformer $\mathsf{TF}_{\theta_n}$ that computes $C_n(x)$ using $T(n)$ steps of CoT for all $x \in \{0,1\}^n$. More precisely, we will construct a transformer that simulates one boolean gate in $C_n$, following the topological order of the circuit, in each step of its chain of thought.

We first index the gates (including input gates) from 1 to $n + T(n)$ according to the topological order. For each gate $i \in [n + 1, n + T(n)]$, we use $a(i)$ and $b(i)$ to denote its two input gates. Since NOT only has one input, we set $a(i)$ as its input and $b(i)$ as 0. We let $c(i) = 0$ if $i$th gate is NOT and $c(i) = 1$ if $i$th gate is AND. For any input gate $1 \leq i \leq n$, $a(i), b(i)$, and the gate type are not meaningful and their choice will not affect the output and thus can be set arbitrarily. For convenience, we will set $a(i) = b(i) = c(i) = 0$. We use $g_i(x)$ to denote the output of non-input gate $i$ ($n + 1 \leq i \leq n + T(n)$) on the circuit input $x \in \{0,1\}^n$, which is equal to $(1 - c(i))(1 - x_{a(i)}) + c(i)(\mathsf{relu}(x_{a(i)} + x_{b(i)} - 1))$.

Now we describe the construction of the vocabulary $\mathcal{V}$, the token embedding $\theta_{\mathsf{TE}}$, and position encoding $\theta_{\mathsf{PE}}$. We set precision $s$ as any positive integer larger than 1, $\mathcal{V} = \{0, 1\}$, $k \triangleq k(n) = \lceil \log_2(T(n) + n) \rceil = O(\log n)$ since $T(n)$ is a polynomial, $d'(n) = 3k + 6$, $\theta_{\mathsf{TE}}(0) = 0 \cdot e_1$, $\theta_{\mathsf{TE}}(1) = 1 \cdot e_1$, and

$$(\theta_{\mathsf{PE}})_{i-1}^\top = [0, 0, 0, 0, c(i), \mathsf{sbin}_k(i), \mathsf{sbin}_k(a(i)), \mathsf{sbin}_k(b(i)), 1], \quad \forall 2 \leq i \leq n + T(n). \quad (6)$$

We use $h_i^0, h_i^{0.5}, h_i^1, h_i^{1.5}$, and $h_i^2$ to denote the intermediate embeddings at position $i$ and different depths. Here, depth 0.5 and 1.5 refer to the output of the Attention layer inside each transformer layer.

1. For the first attention layer, denoting embedding at the $i$th position $h_i^0$ by $h$, we set the query as $q_i \triangleq (h_{k+6:2k+5}) ^\frown (h_{3k+6} \cdot 1_s)$, the key as $k_i \triangleq (B_s h_{6:k+5}) ^\frown (-h_{3k+6} \cdot 1_s)$, and the value as $v_i \triangleq h_1 \cdot e_2$. [7]

2. For the first fully-connected layer, we skip it by setting its weights to be 0.

3. For the second attention layer, denoting embedding at the $i$th position $h_i^1$ by $h$, we set the query as $q_i \triangleq (h_{2k+6:3k+5}) ^\frown (h_{3k+6} \cdot 1_s)$, the key as $k_i \triangleq (B_s h_{6:k+5}) ^\frown (-h_{3k+6} \cdot 1_s)$, and the value as $v_i \triangleq h_1 \cdot e_3$.

4. For the second fully-connected layer, we define $F(a, b, c) \triangleq \mathsf{relu}(1 - a - c) + \mathsf{relu}(a + b + c - 2)$. Denote the embedding at position $i$, $h_{1.5}^i$ by $h$. The output of the second fully-connected layer is defined as $F(h_2, h_3, h_4) \cdot e_4$. Note this expression is valid because it can be expressed by two-layer ReLU nets with constant bits of precision and a constant number of neurons.

5. The final output at position $i$ is $(h_i^2)_4$.

Below we first describe the value of the internal variables of the transformer and then show there exist parameters making such computation realizable. Let $(x_1, \ldots, x_n)$ be the input tokens, we

---

[7] Note here the dimension of $k_i$ and $q_i$ are the same but less than $d'$, which does not strictly satisfy our definition of transformer in Algorithm 3. This is for notational convenience and is without loss of generality because we can pad extra zeros.

define $x_{n+i} \triangleq \mathsf{TF}_{\theta_n}^i(x_1, \ldots, x_n), \forall 1 \leq i \leq T(n)$. We claim there exists transformers parameter $\theta_n$ such that $\mathsf{TF}_{\theta_n}^{T(n)}(x_1, \ldots, x_n) = g_{T(n)}(x)$ (*i.e.* $C_n(x)$). More specifically, we claim that our constructions will ensure the following inductively for all $n + 1 \leq i \leq n + T(n)$,

1. $h_{i-1}^0 = [x_{i-1}, 0, 0, 0, c(i), (\mathsf{sbin}_k(i))^\top, (\mathsf{sbin}_k(a(i)))^\top, (\mathsf{sbin}_k(b(i)))^\top, 1]^\top$;

2. $h_{i-1}^{0.5} = [x_{i-1}, x_{a(i)}, 0, 0, c(i), (\mathsf{sbin}_k(i))^\top, (\mathsf{sbin}_k(a(i)))^\top, (\mathsf{sbin}_k(b(i)))^\top, 1]^\top$;

3. $h_{i-1}^1 = [x_{i-1}, x_{a(i)}, 0, 0, c(i), (\mathsf{sbin}_k(i))^\top, (\mathsf{sbin}_k(a(i)))^\top, (\mathsf{sbin}_k(b(i)))^\top, 1]^\top$;

4. $h_{i-1}^{1.5} = [x_{i-1}, x_{a(i)}, x_{b(i)}, 0, c(i), (\mathsf{sbin}_k(i))^\top, (\mathsf{sbin}_k(a(i)))^\top, (\mathsf{sbin}_k(b(i)))^\top, 1]^\top$;

5. $h_{i-1}^2 = [x_{i-1}, x_{a(i)}, x_{b(i)}, g_i(x), c(i), (\mathsf{sbin}_k(i))^\top, (\mathsf{sbin}_k(a(i)))^\top, (\mathsf{sbin}_k(b(i)))^\top, 1]^\top$;

6. $x_i \triangleq \mathsf{TF}_{\theta_n}(x_1, \ldots, x_n, \ldots, x_{i-1}) = g_i(x)$.

Now we explain why the above conditions hold for any position $i$ using induction, *i.e.*, assuming it is true for all $n + 1 \leq i' \leq i$. We first notice that by our construction, for all $1 \leq i' \leq n - 1$, it holds that $(h_{i'}^k)_1 = x_{i'}$ and $(h_{i'}^k)_{6:k+5}$ for all $k \in \{0, 0.5, 1, 1.5, 2\}$. Note these are the only information that will be used in the later attention layers.

1. This is simply by the construction of $\theta_{\mathsf{TE}}$ and $\theta_{\mathsf{PE}}$.

2. In the first attention layer, at the $j$th position, we have $q_j \triangleq \mathsf{sbin}_k(a(j))^\frown 1_s$ as the query, $k_j \triangleq B_s \cdot \mathsf{sbin}_k(j)^\frown(-1_s)$ as the key and $v_j \triangleq x_j$ as the value for all $j \in [n + T(n)]$. Note here we reduce the sizes of hidden embeddings for simplicity of demonstration. This is valid because we can fill the extra coordinates by 0. This is valid because we can always set the extra coordinates to be 0. By Lemma F.3, we know that $\left[\exp(\langle q_i, k_j \rangle_s)\right]_s = \mathbf{1}[a(i) = j]$ for all $j \in [n + T(n)]$. Recall that the attention scores are defined as $s_i \triangleq \mathsf{softmax}(\langle q_i, k_1 \rangle_s, \ldots, \langle q_i, k_i \rangle_s) \| (0, \ldots, 0)$, we know that $s_i = e_{a(i)}$.

3. We set the parameters in the first fully-connected feedforward layer to be all 0 and let the skip connection pass the intermediate values.

4. The second attention layer attains $x_{b(i)}$ and places it in the third coordinate in the same way as step 2.

5. In the fully-connected feedforward layer we compute $F(x_{a(i)}, x_{b(i)}, c(i)) = \mathsf{relu}(1 - x_{a(i)} - c(i)) + \mathsf{relu}(x_{a(i)} + x_{b(i)} + c(i) - 2)$ for all $x_{a(i)}, x_{b(i)}, c(i) \in \{0, 1\}$. We can verify that $F(x_{a(i)}, x_{b(i)}, c(i)) = (1 - c(i))(1 - x_{a(i)}) + c(i)(\mathsf{relu}(x_{a(i)} + x_{b(i)} - 1)) = g_i(x)$, which is the desirrd output of the gate $i$. This is because when $c(i) = 0$, the output is $1 - a(i) = \mathsf{NOT}a(i)$ and when $c(i) = 0$, the output is $\mathsf{relu}(a(i) + b(i) - 1) = a(i)\mathsf{AND}b(i)$.

6. The output layer uses the fourth coordinate of $h_i^2$, which is $g_i(x)$ according to induction, as the output.

This completes the proof of Theorem 3.3. $\qquad\square$

## F.2 PROOF OF THEOREMS 3.7 AND 3.8

In this subsection, we prove Theorems 3.7 and 3.8. We first prove a useful lemma that gives an equivalent characterization of $\mathsf{SIZE}^{\mathsf{AC}^0}$ and $\mathsf{SIZE}^{\mathsf{TC}^0}$.

**Lemma F.4.** For any $T(n) \in \mathsf{poly}(n)$ satisfying $T(n) \geq 1$, $\forall n \in \mathbb{N}^+$, a decision problem $\mathcal{L} : \cup_{k \in \mathbb{N}^+} \{0, 1\}^k \to \{0, 1\}$ belongs to $\mathsf{SIZE}^{\mathsf{AC}^0}[T(n)]$ (resp. $\mathsf{SIZE}^{\mathsf{TC}^0}[T(n)]$) if and only if there exist a polynomial $S(n)$, a function $T'(n) = O(T(n))$, and a depth $L \in \mathbb{N}^+$ such that for every $n \in \mathbb{N}^+$ there exist a sequence of sizes-$S(n)$, depth-$L$ circuits, $\{C_n^i\}_{i=1}^{T'(n)}$, with unlimited-fanin AND, OR

and NOT gates (with additionally MAJORITY gates for $\mathsf{SIZE}^{\mathsf{TC}^0}$) and that for all $x \in \{0,1\}^n$,

$$\mathcal{L}(x) = x_{n+T'(n)}, \quad \text{where} \quad x_{n+i} \triangleq C_n^i(x_1,\ldots,x_{n+i-1}), \quad \forall i \in [T'(n)]. \tag{7}$$

*Proof of Lemma F.4.* We will prove for $\mathsf{SIZE}^{\mathsf{TC}^0}$ only and the proof for $\mathsf{SIZE}^{\mathsf{AC}^0}$ is almost the same.

The " $\implies$ " direction is straightforward. By definition of $\mathsf{SIZE}^{\mathsf{TC}^0}[T(n)]$ (Definition 3.5), for any $\mathcal{L} \in \mathsf{SIZE}^{\mathsf{TC}^0}[T(n)]$, there is a function $p(n) \in \mathsf{poly}(n)$ and a family of $\mathsf{TC}^0$ circuits $\{C_i'\}_{i=1}^\infty$ such that for every $n \in \mathbb{N}$ and $x \in \{0,1\}^n$, $\mathcal{L}_n(x_1,\ldots,x_n)$ can be computed by a size-$O(T(n))$ threshold circuits with oracle gate $C_{p(n)}$. Now we sort all the nodes in the circuits with oracle gates along the topological order as $x_1,\ldots,x_{n+T'(n)}$ where $x_1,\ldots,x_n$ are the inputs and $T'(n) = O(T(n))$ is the number of the gates, then clearly $x_{n+i}$ is a function of $x_1,\ldots,x_{n+i-1}$ for all $i \in [T'(n)]$ and this function can be implemented by a different threshold circuit of constant depth and $\mathsf{poly}(n)$ size for each $i$. This completes the proof of " $\implies$ " direction.

Now we prove the other direction "$\impliedby$". We first show that given $T'(n)$ sizes-$S(n)$, depth-$L$ circuits, $\{C_n^i\}_{i=1}^{T'(n)}$, there is a depth-$(L+1)$, size $O(T'(n)S(n))$ circuit $C_n'$, such that

$$C_n'(x_1,\ldots,x_{n+T'(n)-1}, e_j) = C_n^j(x_1,\ldots,x_{n+j-1}), \quad \forall j \in [T'(n)], x \in \{0,1\}^{n+T'(n)-1}, \tag{8}$$

where $\mathbf{1}_j \in \{0,1\}^{T'(n)}$ is the one-hot vector with its $j$th coordinate being 1. Indeed, it suffices to set

$$C_n'(1,\ldots,x_{n+T'(n)-1}, y_1,\ldots,y_{T'(n)}) = \vee_{j=1}^{T'(n)} \left( y_j \wedge C_n^j(x_1,\ldots,x_{n+j-1}) \right). \tag{9}$$

Once we have such oracle gate $C_n'$, given input $x_1,\ldots,x_n$, we can recursively define

$$x_{n+i} \triangleq C_n'(x_1,\ldots,x_{n+i-1}, 0_{T'(n)-n}, e_j). \tag{10}$$

Thus we can compute $\mathcal{L}(x) = x_{n+T'(n)}$ using $T'(n)$ oracle gate $C_n'$. We can get constant gate 0 and 1 by using $x_1 \wedge \neg x_1$ and $x_1 \vee \neg x_1$. respectively. This completes the proof. $\quad\square$

Now we are ready to prove Theorems 3.7 and 3.8. We will prove Theorem 3.7 first and the proof of Theorem 3.8 is very similar to Theorem 3.7.

*Proof of Theorem 3.7.* We first show that $\mathsf{SIZE}^{\mathsf{TC}^0}[T(n) + 1] \supseteq \mathsf{CoT}[T(n), \mathsf{poly}(n), \log n]$. For the case that the vocabulary of transformer $\mathcal{V} = \{0,1\}$, by Theorem 3.2, we know for any $\theta_n$, $\mathsf{TF}_{\theta_n}(x_1,\ldots,x_i)$ can be expressed by a $\mathsf{TC}^0$ circuit whose depth is uniformly upper bounded by some constant for all $n \leq i \leq n + O(T(n))$. This completes the proof when $\mathcal{V} = \{0,1\}$. When $\mathcal{V} \neq \{0,1\}$, we can use the binary encoding of elements in $\mathcal{V}$ as the inputs of those $\mathsf{TC}^0$ gates constructed for the later layers of the transformer.

Now we turn to the proof for the other direction: $\mathsf{SIZE}^{\mathsf{TC}^0}[T(n) + 1] \subseteq \mathsf{CoT}[T(n), \mathsf{poly}(n), \log n]$. In high-level speaking, the proof contains two steps:

1. We show that $\mathsf{TC}^0 \subseteq \mathsf{T}[\mathsf{poly}(n), \log n] \subseteq \mathsf{CoT}[1, \mathsf{poly}(n), \log n]$. The first step has two key constructions: (a). using attention to copy all the weights to the same position; (b). we can use polysize two-layer FC net with ReLU activation to simulate MAJORITY, AND, OR gate with unbounded fan-in (Lemma F.5);

2. We can do the first step for all positions $i = n+1,\ldots,n+O(T(n)+1)$ simultaneously.

By Lemma F.4, we know that for any problem $\mathcal{L} \in \mathsf{SIZE}^{\mathsf{TC}^0}[T(n) + 1]$, there exist constant $L$, polynomial $S(n)$, and $T'(n) = O(T(n) + 1)$, such that for every $n \in \mathbb{N}^+$, there exist a sequence of threshold circuits, $\{C_n^i\}_{i=1}^{T'(n)}$, whose sizes are uniformly bounded by $S(n)$ and depth are uniformly bounded by $L$, and that for all $x \in \{0,1\}^n$,

$$L(x) = x_{n+T'(n)}, \quad \text{where} \quad x_{n+i} \triangleq C_n^i(x_1,\ldots,x_{n+i-1}), \quad \forall i \in [T'(n)]. \tag{11}$$

To simplify the notation of the proof, without loss of generality, we assume for each $i \in [T'(n)]$, circuit $C_n^i$ has the exactly same size $S(n)$ and depth $L$.

Now we present the construction of the constant-depth, constant-precision decoder-only transformer, $\mathsf{TF}_{\theta_n}$ which computes problem $\mathcal{L}$ when input length is $n$. Without loss of generality we only consider the case where $T'(n) = T(n) + 1$. We set vocabulary $\mathcal{V} = \{0, 1\}$, embedding width $d(n) = 1 + 3(T(n) + n) + 2S(n)T(n) = O(\mathsf{poly}(n))$, depth equal to $L + 2$, CoT length $T(n)$ and precision $s(n) = \lceil \log_2 S(n) \rceil$ so the precision is high enough for simulating all the $\mathsf{poly}(n)$ size MAJORITY gates used in $C_n^i$ (Lemma F.5). We set $(\theta_{\mathsf{TE}})_0 = 0 \cdot e_1$, $(\theta_{\mathsf{TE}})_1 = 1 \cdot e_1$, and $(\theta_{\mathsf{PE}})_i = e_{i+1}$ for all $i \in [n + T(n)]$, where we use $e_i \in \{0, 1\}^{d(n)}$ to denote the one-hot vector whose $i$th coordinate is 1 for $i \in [d(n)]$ and $\overline{e}_i \in \{0, 1\}^{n+T(n)}$ to denote one-hot vector whose $i$th coordinate is 1 for $i \in [n + T(n)]$.

Below we first describe the value the internal variables of the transformer and then show there exist parameters making such computation realizable. To make our claims more interpretable, we only write the non-zero part of the embedding and omit the remaining 0's. the Let $(x_1, \ldots, x_n)$ be the input tokens and $\Delta \triangleq 3n + 3T(n) + 1$, our constructions will ensure that

1. $x_{n+i} = \mathsf{TF}_{\theta_n}^i(x_1, \ldots, x_n), \forall i \in [T(n)]$.

2. $h_i^0 = x_i e_1 + e_{i+1} = (x_i, \overline{e}_i) \, \forall i \in [n + T(n)]$;

3. $h_i^{0.5} = x_i e_1 + e_{i+1} = (x_i, \overline{e}_i), \, \forall i \in [n + T(n)]$;

4. $h_i^1 = x_i e_1 + e_{i+1} + x_i e_{n+T(n)+i+1} = (x_i, \overline{e}_i, x_i \overline{e}_i), \, \forall i \in [n + T(n)]$;

5. $h_i^{1.5} = (x_i, \overline{e}_i, x_i \overline{e}_i, x_1, \ldots, x_i), \forall i \in [n + T(n)]$

6. $(h_{n+i-1}^{1+l})_{\Delta+(i-1)S(n)+1:\Delta+iS(n)} \in \{0, 1\}^{S(n)}$ stores the intermediate result of circuit $C_n^i(x_1, \ldots, x_{n+i-1})$ at layer $l, \forall i \in [T(n)]$ and $l \in [L]$;

7. $(h_{n+i-1}^{L+2})_{\Delta+(i-1)S(n)+1:\Delta+iS(n)}$ is the intermediate result of circuit $C_n^i(x_1, \ldots, x_{n+i-1})$ at layer $l$, $\forall i \in [T(n)]$, but represented using $\{-1, 1\}$. That is, $(h_{n+i-1}^{L+2})_{\Delta+(i-1)S(n)+1:\Delta+iS(n)} = 2 \cdot (h_{n+i-1}^{L+2})_{\Delta+(i-1)S(n)+1:\Delta+iS(n)} - 1$. Meanwhile, $(h_{n+i-1}^{L+2})_{\Delta+(j-1)S(n)+1:\Delta+jS(n)} = 0, \forall i, j \in [T(n)], j \neq i$

8. $(\theta_{\mathsf{OUTPUT}} h_{n+i-1}^{L+2})_0 = 0, (\theta_{\mathsf{OUTPUT}} h_{n+i-1}^{L+2})_1 = \sum_{j=1}^{T(n)} (h_{n+i-1}^{L+2})_{\Delta+jS(n)} = 2C_n^i(x_1, \ldots, x_{n-i+1}) - 1$, for all $i \in [T(n)]$.

Now we explain the purpose of each layer and how to set the parameters such that the requirements above are met.

1. $x_{n+i} = \mathsf{TF}_{\theta_n}^i(x_1, \ldots, x_n), \forall i \in [T(n)]$ is the goal of the construction;

2. This is by our construction of $\theta_{\mathsf{PE}}$ and $\theta_{\mathsf{TE}}$;

3. The first attention layer does nothing by setting all weights to 0;

4. By Lemma F.5, AND can be simulated by 2-layer ReLU networks using 2 hidden neurons. Thus we use the first feedforward-layer to compute the function $(h_i^1)_{n+T(n)+1+j} = (h_i^{0.5})_{1+j} \wedge (h_i^{0.5})_1$ for all $i, j \in [n + T(n)]$ with totally $2(n + T(n))$ hidden neurons. Therefore if $j \neq i$, then $(h_i^{0.5})_{1+j} = 0$, which implies $(h_i^1)_{n+T(n)+1+j} = 0$; if $j = i$, then $(h_i^{0.5})_{1+j} = 1$, thus $(h_i^1)_{n+T(n)+1+j}(h_i^{0.5})_1 = x_i$.

5. This step exactly requires $(\mathsf{ATTN}_{\theta_{\mathsf{ATTN}}^{(1)}}(h_1^{(1)}, \ldots, h_n^{(1)}))_i = \sum_{j=1}^i e_{\Delta+j} x_j$. It suffices to set the attention score of the second layer at $i$th position $s_i = (1, \ldots, 1, 0 \ldots, 0) = (1_i, 0_{n+T(n)-i})$ for all $i \in [n + T(n) - 1]$. This can be done by setting $q_i^{1.5} = W_Q h_i^{1.5} = (B_{s(n)}, 0_{n+T(n)-1}), k_i^{1.5} = W_K h_i^{1.5} = (1, 0_{n+T(n)-1})$. By Lemma F.2, we

have $[\exp([\langle q_i, k_j \rangle]_{s(n)})]_{s(n)} = [\exp(B_{s(n)})]_{s(n)} = B_{s(n)}$. Since rounded sum of any number of $B_{s(n)}$ is still $B_{s(n)}$ and $[B_{s(n)}/B_{s(n)}]_{s(n)} = 1$, we know that

$$s_i = \mathrm{softmax}_{s(n)}(B_{s(n)}1_i)\|(0,\ldots,0) = (1,\ldots,1,0\ldots,0) = (1_i, 0_{n+T(n)-i})$$

for all $i \in [n + T(n) - 1]$. Note in this step we use our specific rounding rule to copy all the previous $x_i$ with a sum of attention score larger than 1. We can just also use approximately uniform attention scores with an additional coefficient before $x_i$ since we have $\log n$ precision. Finally we set $v_i^{1.5} = W_V h_i^{1.5} = e_{\Delta+i}x_i$ and $W_O = I_{d(n)}$.

6. The second attention layer is the only attention layer which has non-zero weights. Using the feedforward ReLU networks from layer 2 to $L + 1$, we can simulate the circuits $C_n^i$ in parallel for all $i \in [T(n)]$ by Lemma F.5. In detail, Lemma F.5 ensures that we can use a two-layer fully-connected ReLU network with weights to simulate a layer of the $\mathsf{TC}^0$ circuits $C_n^i$. Moreover, there is enough space in the embedding to reserve $S(n)$'s 1 needed by Lemma F.5. And we always append the value of intermediate gates after the gate values which have already been computed at each layer using the indices $\Delta + (i-1)S(n) + 1 : \Delta + S(n)$ for $h_{n+i-1}^l$ for each $i \in [T(n)]$. Note for each $i$, only the computation in the range $\Delta + (i-1)S(n) + 1 : \Delta + S(n)$ is meaningful and the computation for other indices will not be used later.

7. This is similar to step 3. We skip the attention layer and simply set $(h_{n+i-1}^{L+2})_{\Delta+(i-1)S(n)+k} = (h_{n+i-1}^{L+1.5})_{n+i} \wedge (h_{n+i-1}^{L+1.5})_{\Delta+(i-1)S(n)+k}$ for all $k \in [S(n)]$ using the feedforward fully-connected network.

8. This step holds directly due to the property guaranteed in step 8. We note that with the property claimed in step 9, we have that $(\theta_{\mathsf{OUTPUT}}h_{n+i-1}^{L+2})_1 - (\theta_{\mathsf{OUTPUT}}h_{n+i-1}^{L+2})_0 = 2C_n^i(x_1,\ldots,x_{n-i+1})) - 1$. Thus if $C_n^i(x_1,\ldots,x_{n-i+1}) = 1$, then $(\theta_{\mathsf{OUTPUT}}h_{n+i-1}^{L+2})_1 - (\theta_{\mathsf{OUTPUT}}h_{n+i-1}^{L+2})_0 = 1$, which implies $\mathsf{TF}_{\theta_n}(x_1,\ldots,x_{n+i-1}) = 1$, otherwise if $C_n^i(x_1,\ldots,x_{n-i+1}) = 0$, then $\mathsf{TF}_{\theta_n}(x_1,\ldots,x_{n+i-1}) = 0$. In both cases, we have that

$$C_n^i(x_1,\ldots,x_{n-i+1}) = \mathsf{TF}_{\theta_n}(x_1,\ldots,x_{n+i-1}) \tag{12}$$

So far we have finished the proof for the general $T(n)$. Specifically, when $T(n) = T'(n) = 0$, our proof shows that the constant-depth transformer can still simulate any constant-depth circuit, which means $\mathsf{TC}^0 \subseteq \mathsf{T}[\mathsf{poly}(n)] \subseteq \mathsf{CoT}[1, \mathsf{poly}(n)] = \mathsf{SIZE}^{TC^0}(1) = \mathsf{TC}^0$. Thus all the inclusions are equivalence, that is $\mathsf{TC}^0 = \mathsf{T}[\mathsf{poly}(n)] = \mathsf{CoT}[1, \mathsf{poly}(n)] = \mathsf{SIZE}^{TC^0}(1)$. $\qquad\square$

*Proof of Theorem 3.8.* We first show that $\mathsf{SIZE}^{\mathsf{AC}^0}[T(n) + 1] \supseteq \mathsf{CoT}[T(n), \mathsf{poly}(n), 1]$. For the case that the vocabulary of transformer $\mathcal{V} = \{0, 1\}$, by Theorem 3.1, we know for any $\theta_n$, $\mathsf{TF}_{\theta_n}(x_1,\ldots,x_i)$ can be expressed by a $\mathsf{TC}^0$ circuit whose depth is uniformly upper bounded by some constant for all $n \le i \le n + O(T(n))$. This completes the proof when $\mathcal{V} = \{0, 1\}$. When $\mathcal{V} \ne \{0, 1\}$, we can use the binary encoding of elements in $\mathcal{V}$ as the inputs of those $\mathsf{TC}^0$ gates constructed for the later layers of the transformer.

The other direction is almost the same as that of Theorem 3.7, except that we now only need constant bits of precision because we do not need to simulate MAJORITY gates (Lemma F.5).

$\qquad\square$

## F.3 PROOF OF THEOREM 3.9

*Proof of Theorem 3.9.* By Lemma F.7, it holds that for all $k \in \mathbb{N}$, $\mathsf{AC}^0 \subsetneq \mathsf{SIZE}[n^k]$. By Theorem 3.7, we know that $\mathsf{AC}^0 \subseteq \mathsf{CoT}[1, \mathsf{poly}(n), 1] \subseteq \mathsf{CoT}[n^k, \mathsf{poly}(n), 1]$ for any $k \in \mathbb{N}$. Thus $\mathsf{CoT}[n^k, \mathsf{poly}(n), 1] \subsetneq \mathsf{SIZE}[n^k]$ for all $k, k' \in \mathbb{N}$. Also, note that the attention layer and fully-connected layer can be computed using poly-size circuits. Thus for any $k \in \mathbb{N}$, $\mathsf{CoT}[n^k, \log(n)] \subseteq \mathsf{SIZE}[n^{k'}]$ for some integer $k' \ge k$. Combining these we conclude that for any $k \in \mathbb{N}$, $\mathsf{CoT}[n^k, \log(n)] \subsetneq \mathsf{CoT}[n^k, \mathsf{poly}(n)]$. $\qquad\square$

### F.4 AUXILIARY LEMMAS

In this subsection, we prove a few auxiliary lemmas that are used in the proofs in Section 3.4.

**Lemma F.5.** Unlimited-fanin $\mathsf{AND}, \mathsf{OR}$ (resp. $\mathsf{MAJORITY}$) : $\{0,1\}^n \to \{0,1\}$ can be simulated by some 2-layer feedforward ReLU network with constant (resp. $\log n$) bits of precision constant hidden dimension and additional $n$ constant inputs of value 1.

Mathematically, let $\mathsf{FF}[s(n)]$ be the set of functions $C : \{0,1\}^n \to \{0,1\}$ which can be a two-layer feedforward ReLU network with at most $s(n)$ bits of precision and constant hidden dimension $\mathsf{FF}_\theta : \{0,1\}^{2n} \to \{0,1\}, \mathsf{FF}_\theta(x') = W_2 \times_s \mathsf{relu}([W_1 \times_s x' + b_1]_s)$, where $\theta = (W_2, W_1, b_1)$, such that for any $x \in \{0,1\}^n$,

$$\mathsf{FF}_\theta(x_1, 1, x_2, 1, \ldots, x_n, 1) = C(x). \tag{13}$$

We have unlimited-fanin $\mathsf{AND}, \mathsf{OR} \in \mathsf{FF}[1]$ and $\mathsf{MAJORITY} \in \mathsf{FF}[\log n]$.

The proof of Lemma F.5 is based on the following straightforward lemma (Lemma F.6).

**Lemma F.6.** For any $e \in \mathbb{N}^+$ and $a \in \mathbb{Z} \cap \mathbb{F}_s$, $\mathsf{relu}([a]_s) - \mathsf{relu}([a-1]_s) = \mathbb{1}[a > 0]$. In particular, for any $a \in \mathbb{Z}$, $\mathsf{relu}(a) - \mathsf{relu}(a-1) = \mathbb{1}[a > 0]$.

*Proof of Lemma F.5.* Recall that $x^\frown y$ denotes $(x_1, y_1, x_2, y_2, \ldots, x_n, y_n)$ for any $x, y \in \{0,1\}^n$. We have that $\mathsf{sum}_s(x^\frown(-1_n)) \le 0$ for all $e \ge 2$ and $x \in \{0,1\}^n$. Moreover, $\mathsf{sum}_s(x^\frown(-1_n)) = 0 \iff \forall i \in [n], x_i = 1$. Similarly, we have that $\mathsf{sum}_s(x) \ge 0$ and $\mathsf{sum}_s(x) = 0 \iff \forall i \in [n], x_i = 0$. In other words, we have

- $\mathsf{AND}(x) = \mathbb{1}[\mathsf{sum}_s(x^\frown(-1_n)] \ge 0) = \mathbb{1}[\langle x^\frown 1_n, 1_n^\frown(-1_n)\rangle_s + 1 > 0]$;
- $\mathsf{OR}(x) = \mathbb{1}[\mathsf{sum}_s(x_i') > 0] = \mathbb{1}[\langle x^\frown 1_n, 1_n^\frown(0_n)\rangle_s > 0]$.

Therefore for AND, we can set $\theta^{\mathsf{AND}} \triangleq (W_1^{\mathsf{AND}}, W_2^{\mathsf{AND}}, b_1^{\mathsf{AND}})$ with $W_1^{\mathsf{AND}} \triangleq \begin{bmatrix} 1_n^\frown(-1_n) \\ 1_n^\frown(-1_n) \end{bmatrix}, b_1 = \begin{bmatrix} 1 \\ 0 \end{bmatrix}, W_2^{\mathsf{AND}} = [1, -1]$, and we have that

$$
\begin{aligned}
\mathsf{FF}_{\theta^{\mathsf{AND}}}(x^\frown 1_n) &= [\mathsf{relu}(\langle x^\frown 1_n, 1_n^\frown(-1_n)\rangle_s + 1) - \mathsf{relu}(\langle x^\frown 1_n, 1_n^\frown(-1_n)\rangle_s)]_s \\
&= \mathbb{1}[\langle x^\frown 1_n, 1_n^\frown(-1_n)\rangle_s + 1 > 0] \qquad \text{(by Lemma F.6)} \\
&= \mathsf{AND}(x)
\end{aligned}
$$

Similarly for OR, we can set $\theta^{\mathsf{OR}} \triangleq (W_1^{\mathsf{OR}}, W_2^{\mathsf{OR}}, b_1^{\mathsf{OR}})$ with $W_1^{\mathsf{OR}} \triangleq \begin{bmatrix} 1_n^\frown 0_n \\ 1_n^\frown 0_n \end{bmatrix}, b_1 = \begin{bmatrix} 0 \\ -1 \end{bmatrix}, W_2^{\mathsf{OR}} = [1, -1]$, and we have that

$$
\begin{aligned}
\mathsf{FF}_{\theta^{\mathsf{OR}}}(x^\frown 1_n) &= [\mathsf{relu}(\langle x^\frown 1_n, 1_n^\frown 0_n\rangle_s) - \mathsf{relu}(\langle x^\frown 1_n, 1_n^\frown 0_n\rangle_s - 1)]_s \\
&= \mathbb{1}[\langle x^\frown 1_n, 1_n^\frown 0_n\rangle_s > 0] \qquad \text{(by Lemma F.6)} \\
&= \mathsf{OR}(x)
\end{aligned}
$$

The proofs for AND and OR are thus completed.

Next we deal with MAJORITY. Note that for $s(n) \ge \log_2 n + 1$, we have that $\sum_{i=1}^n (2x_i - 1) = \langle x^\frown 1_n, 2_n^\frown(-1_n)\rangle_s$ for all $x \in \{0,1\}^n$.

$$
\begin{aligned}
\mathsf{MAJORITY}(x) &= \mathbb{1}[\sum_{i=1}^n (2x_i - 1) > 0] = \mathbb{1}[\langle x^\frown 1_n, 2_n^\frown(-1_n)\rangle_s > 0] \\
&= [\mathsf{relu}(\langle x^\frown 1_n, 2_n^\frown(-1_n)\rangle_s) - \mathsf{relu}(\langle x^\frown 1_n, 2_n^\frown(-1_n)\rangle_s - 1)]_s \\
&= \mathsf{FF}_{\theta^{\mathsf{MAJORITY}}}(x^\frown 1_n), \tag{14}
\end{aligned}
$$

where $\theta^{\mathsf{MAJORITY}} \triangleq (W_1^{\mathsf{MAJORITY}}, W_2^{\mathsf{MAJORITY}}, b_1^{\mathsf{MAJORITY}})$ with $W_1^{\mathsf{MAJORITY}} \triangleq \begin{bmatrix} 2_n^\frown -1_n \\ 2_n^\frown -1_n \end{bmatrix}, b_1 = \begin{bmatrix} 0 \\ -1 \end{bmatrix}, W_2^{\mathsf{MAJORITY}} = [1, -1]$. $\qquad \square$

**Lemma F.7.** *For all $k \in \mathbb{N}$, $\mathsf{AC}^0 \not\subseteq \mathsf{SIZE}[n^k]$.*

*Proof of Lemma F.7.* We first define $\overline{\mathsf{SIZE}}[T(n)]$ as the problems solvable by circuits with $T(n)$ standard $\mathsf{AND}, \mathsf{OR}, \mathsf{NOT}$ gates exactly. Thus $\mathsf{SIZE}[n^k] = \cup_{C \in \mathbb{N}^+} \overline{\mathsf{SIZE}}(Cn^k)$. Now we claim that for each $C \in \mathbb{N}$, there is a $N \in \mathbb{N}^+$, such that for all $n \geq N$, it holds that there is a conjunction normal form (CNF) with at most $n^{k+1}$ clauses over $\{x_1, \ldots, x_n\}$ that cannot be expressed by any circuit of size $Cn^k$. This claim holds because of a simple counting argument. There are at least $2^{n^{k+1}}$ different such CNFs. On the other hand, it is well known that one can represent a $T(n)$-size circuit only allowing standard $\mathsf{AND}, \mathsf{NOT}, \mathsf{OR}$ gates with $3T(n) \log T(n)$ bits (we need $\log T(n)$ bits to encode the id of a gate). Thus the total number of different circuits of size at most $Cn^k$ is at most $2^{3Cn^k(k \log n + C)}$, which is smaller than $2^{n^{k+1}}$ for sufficiently large $n$. We denote such $n$ for each $C$ by $N_C$. Now we define the following language $\mathcal{L}_{\mathsf{CNF}}$: if the input length of $x$ is $N_C$ for some $C$, use the $n^{k+1}$-clause CNF's output which cannot be expressed by size-$Cn^k$ circuits as the output; otherwise rejects (output 0). Then clearly $\mathcal{L}_{\mathsf{CNF}} \notin \overline{\mathsf{SIZE}}(Cn^k)$ for all $C$, thus $\mathcal{L}_{\mathsf{CNF}} \notin \cup_{C \in \mathbb{N}^+} \overline{\mathsf{SIZE}}(Cn^k) = \mathsf{SIZE}[n^k]$. By construction, $\mathcal{L}_{\mathsf{CNF}} \in \mathsf{AC}^0$. This completes the proof. $\qquad\square$

# G DISCUSSION ON VARIANTS IN TRANSFORMER ARCHITECTURE

## G.1 EXTENSION TO TRANSFORMERS WITH LAYERNORM

Allowing LayerNorm changes the function class that a transformer can express and the position of the layer norm also matters (Xiong et al., 2020). However, the expressiveness results mentioned in this work still hold for the two most popular transformer architecture variants with LayerNorm — Post LayerNorm and Pre LayerNorm. The upper bounds on transformer expressiveness Theorems 3.1 and 3.2 clearly don't get affected by adding LayerNorm, which can be computed in polynomial time for each token.

Below we focus on the upper bound of the expressiveness of decoder-only transformers with or without CoT. In detail, we will explain why Theorems 3.3 and 3.7 still holds even with LayerNorm. Here the key observation is that, if each coordinate of $h \in \mathbb{R}^d$ ranges from $\{-1, 1\}$ and $-1, 1$ appear in pairs, then $\mathsf{LayerNorm}(h) = h$. Thus it suffices to show that we can slightly twist the construction of transformers in Theorems 3.3 and 3.7 that for all $i \in [n + T(n)], l \in \{0, 0.5, 1, \ldots, L\}$, $h_i^l$ is composed of $-1$ and $1$ and they appear in pairs so the sum is always 0. Note that in the current construction, each $h_i^l$ only contains $0, -1, 1$. It suffices to replace each dimension with four dimensions, in the sense $0 \rightarrow (1, -1, 1, -1)$, $1 \rightarrow (1, 1, -1, -1)$ and $-1 \rightarrow (-1, -1, 1, 1)$. This can be done by changing the weights of the token embedding, position encoding, and the weights of the second layer of each fully-connected layer. For the outgoing layer, we just use the average of the new representation, which is exactly the same as the original value in all three cases.

## G.2 EXTENSION TO TRANSFORMERS WITH MULTIHEAD ATTENTION

In this paper, for simplicity, we only focus on the case where there is only one attention head in each layer. The main results in this paper still apply if we allow constantly many attention heads, because we can simulate an attention layer with $k$ heads with $k$ attention layers with one head. Allowing an arbitrary number of attention heads while fixing total embedding size might make the constant-depth transformers strictly more expressive in certain settings and we leave it for future works.

# H DISCUSSION ON NON-UNIFORMITY

*Non-uniform* computation models allow a different program for each different input length, like boolean circuits. However, the complexity class defined by circuits can also be uniform, if we add additional assumption on the correlation between circuits of different input lengths, *e.g.*, one can require the circuits for input length $n$ can be generated by a Turing Machine taken $n$ as input in using a certain amount of time and space.

The complexity class $\mathsf{CoT}$ introduced in this paper can also be made uniform by enforcing an additional assumption, that the parameters of the transformer can be generalized by some Turing Ma-

chine given the input sequence length $n$. It is well-known that one can simulate the execution of the Turing Machine for any $T$ steps by a family of uniform boolean circuits of size $O(T^2)$. Thus if we enforce the parameters of transformers in CoT to be uniform, our main theorem would imply that constant-depth transformers with uniform parameters and polynomially many steps of chain of thoughts can solve all problems in P. Also note that the inference of transformers can also be done in polynomial time, we conclude it is exactly equal to P.

One natural question about non-uniformity is that *whether having a different transformer for each input sequence length is practical,* given that a significant portion of previous theoretical works on transformer expressiveness focuses on the uniform setting. This problem is kind of ill-defined because we haven't been able to scale up the input length to arbitrary length in practice, and thus it is not clear if it is necessary to keep scaling up the size of LLMs for longer input sequence length. But at least for the LLMs that have been seen in practice, it seems quite common to scale up the model size when dealing with longer input sequence length. Also taking the GPT architecture (Radford et al., 2019) that we focus on in this paper, having more trainable parameters is necessary for longer input sequence length, due to the trainable absolute position encoding.

Still, one needs to note that there is a difference between natural language tasks and complexity class, where the former has a lot of memorization and does not require a strong ability to solve math problems of any sequence length. In contrast, to learn this complexity class like the composition of permutation of any length, transformers need to have the ability of *length generalization*, which does seem impossible for certain non-uniform models, *e.g.*, like GPT architectures with trainable absolute position encoding, because there is no way to learn the position encoding at an unseen position in the training dataset. Of course, length generalization would still be possible if GPT architecture learned the ground truth without using the trainable position encoding at all.

