# OpenReview forum: "Chain of Thought Empowers Transformers to Solve Inherently Serial Problems"
_ICLR.cc/2024/Conference — ICLR 2024 poster_

### Official Review · Reviewer_woBt · 2023-10-31

**Soundness:** 4 excellent
**Presentation:** 3 good
**Contribution:** 3 good
**Rating:** 8
**Confidence:** 5

**Summary:**

The paper makes several contributions to the theory of the expressive power of transformers via circuit complexity:

1. Constant precision transformers are in $\mathsf{AC}^0$
2. The authors show the existing $\mathsf{TC}^0$ upper bound for log-precision transformers can be extended to iterated addition with intermediate rounding (at least in the case where the exponent has 0 bits).
3. Constant-precision transformers with T steps of CoT and nonuniform $O(\log n)$ embeddings can simulate T-sized circuits and thus express $\mathsf P/\mathsf{poly}$
4. Empirically, transformers get

The first two results relate to understanding the expressive power of encoder-only transformers, and the third shows one sense in which CoT/decoding steps add power to transformers.

**Strengths:**

1. The first contribution (power of hard attention) may not be the most realistic model of practical transformers (see Weaknesses), but it is still potentially valuable for filling out the theory of transformers of different types.
2. The second contribution showing log-precision transformers with rounding are still in TC0 is quite interesting and solves an technical problem unresolved in prior work. Even though the result is not fully general (requires zero exponent), the progress here is quite exciting and perhaps could be extended in future work.
3. I would like some of the assumptions for the P/poly result to be better discussed (see Weaknesses), but it is a valid and potentially useful result for formalizing the power of CoT.
4. The paper is generally well-written and organized, and I appreciate the inclusion of a neat empirical study.

**Weaknesses:**

## Limitations of Constant Precision

Constant precision is not necessarily a realistic setting, since it means transformers cannot attend based on positional encodings or compute uniform attention (which require $\log n$ bits). In the practical regime, transformers have enough precision to express uniform attention over their input length and can use uniform attention to recognize majority ([Merrill et al., 2021](https://aclanthology.org/2022.tacl-1.49/)), which is outside your upper bound of AC0. Presumably, if we wanted to apply transformers on very long input lengths, we would scale up the precision of attention logarithmically so that uniform attention and positional embeddings would remain expressible. For this reason, [Merrill & Sabharwal (2023)](https://neurips.cc/virtual/2023/poster/70153) propose studying the log-precision model instead of the constant precision one.

To put it another way, let's say you ran the same experimental setup as Figure 3 but with Majority instead of Iterated Squaring. If transformers are in AC0, we'd expect a similar qualitative pattern where models without CoT struggle without sufficient depth but models with CoT can succeed. But I think that's not what you would find: instead, even models of one layer could do well using a single uniform attention head.

To be clear, even though I think log-precision is more realistic, I think it is still potentially interesting to analyze the constant-precision case to fill out the overall theory and understand the value of log precision. However, the authors should discuss the differences between constant-precision and log-precision and specifically mention or respond to the argument for log-precision from [Merrill & Sabharwal (2023)](https://neurips.cc/virtual/2023/poster/70153). It could also be helpful to run the experiment I described above with Majority and potentially include the results in the appendix.

## Nonuniform Embeddings

The paper characterizes transformers with T steps as P/poly. There is something weird about this result, in that it characterizes transformers by a nonuniform complexity class that contains undecidable problems (e.g., the unary encoding of the halting problem)! In contrast, we know that transformers cannot compute undecidable problems since they can be implemented on standard computers. This disconnect comes from the fact that the embeddings are assumed to be **nonuniform**: i.e., they can be any sequence of $O(\log n)$ bits on inputs of size $n$. This enables the embeddings to be able to encode advice for solving undecidable problems, which standard positional transformers cannot do because they are computable.

This assumption of nonuniform embeddings should be better highlighted in Section 3.4: right now it's not even visible in the theorem statement. It would also be good to add some discussion of the assumption in the introduction or other high-level parts of the paper so readers don't miss it.

**Questions:**

Your circuit simulation result feels quite related to Lemma 3 from [Merrill & Sabharwal (2023)](https://arxiv.org/pdf/2207.00729.pdf), except that yours uses CoT to avoid the linear depth that they incurred. It could be worth discussing a bit about the similarities and differences between the two constructions.

In Theorems 3.1 and 3.2, you write $\mathsf T[\mathrm{poly}(n), 1, 1] \subseteq \mathsf{CoT}[1, \mathrm{poly}(n), 1, 1]$, where the first class corresponds to $\mathsf{CoT}(0, \ldots)$. Is it possible to say that these classes are actually equal, i.e., a single step of CoT does not add power?

Do you have any examples or thoughts about how transformers might systematically use rounding to gain expressive power (with nonzero exponent)?

G.1: "ranges from {-1, 1}" -> "[-1, 1]"

nit: Wording Problem -> Word Problem?

---

> ### Author Response · Authors · 2023-11-22
> **Response to Reviewer woBt**
>
> **Q1: Constant precision is less practical than log precision.**
>
> >**Response**: We respectfully disagree.
> >
> > First, we would like to highlight that constant precision and log precision are only properly defined for a sequence of language models where the input sequence length goes to infinity. For a concrete and practically used language model, whether it is of constant precision or log precision is an ill-defined question.
> >
> > Still, we can test if the implications or the predictions of certain precision hold in practice, and use it to infer the practicality of that precision. A most notable application of log precision is the ability to perform uniform attention. In detail, given sequence lengths $n$, it requires that the model can express all fractionals, $1,\ldots, 1/n$. Or as a weaker requirement, it requires that the model can express all integers between $1$ and $n$.
> >
> > Below we shall see one of the most successful open-sourced LLM, Llama 2, doesn't have the ability to express uniform attention, or just all natural numbers smaller than its sequence length, 4096. Llama 2 uses BF16, which has 1 sign bit, 8 exponent bits, and 7 precision bits (plus an implicit leading bit). This means $2^8+1=257$ is not expressible in BF16, which is much smaller than the sequence length of Llama 2, 4096. To get more intuition about BF16, we note that 255+2 = 256 in BF16. Nowadays practitioners are even pursuing more extreme low precision models, like this recent work by [Peng et al, 2023], which uses FP8 to language models. Here FP8 contains 1 sign bit, 4 exponent bit, and 3 bits of mantissa.
> >
> > That being said, we also agree with the reviewer that our new AC0 upper bound for constant precision transformers does not imply constant-depth transformers in practice cannot learn problems beyond AC0 without COT. At this time point, it is not clear what architecture and precision transformers will converge to in the future, and thus we believe the study of both precision is meaningful and provides useful insight to the community.
>
> **Q2: Comparison to Lemma 3 from [Merrill & Sabharwal, 2023]**
>
> > **Response:** First the goals are different. We are trying to use a transformer with CoT to simulate an arbitrarily given circuit, while Lemma 3 from [Merrill & Sabharwal, 2023] aims to compute a fixed problem, which is the Circuit Value Problem. Our goal is more general in the sense that the CVP problem itself can be solved by some circuit and we just need to simulate that circuit.
> Second, the results are different. The steps of CoT we need is the number of the gates of the circuit to be simulated, while their depth is an upper bound for the depth of the circuits where the CVP problem can be solved. The major difference here is because the vocabulary is finite and each time though it is possible to compute more in parallel, one step of CoT can only write constant bits of information, instead of the whole layer of the circuit.
>
> **Q3: Is it possible to show  $\mathsf T[\mathrm{poly}(n), 1, 1] = \mathsf{CoT}[1, \mathrm{poly}(n), 1, 1]$?**
>
> > **Response:** Good question. We do not know if it is possible. But Theorem 3.8 and 3.9 suggest that one step of COT does not help if one has $\mathsf{poly}(n)$ embedding size.
>
> **Q4: Do you have any examples or thoughts about how transformers might systematically use rounding to gain expressive power (with nonzero exponent)?**
>
> > **Response:** Excellent question. Unfortunately, we do not have a good answer. We leave it for future work.
>
> **References:**
> - Peng, H., Wu, K., Wei, Y., Zhao, G., Yang, Y., Liu, Z., Xiong, Y., Yang, Z., Ni, B., Hu, J. and Li, R., 2023. FP8-LM: Training FP8 Large Language Models. arXiv preprint arXiv:2310.18313.

---

> > ### Comment · Reviewer_woBt · 2023-11-22
> >
> > Thanks for your response!
> >
> > > Llama 2 uses BF16, which has 1 sign bit, 8 exponent bits, and 7 precision bits (plus an implicit leading bit). This means $2^8 + 1 = 257$ is not expressible in BF16, which is much smaller than the sequence length of Llama 2, 4096.
> >
> > FWIW, this seems off. 2^8 is the maximum representable value with 8 *mantissa* bits, but with 8 exponent bits, we can represent 2^2^8, which is much larger than 4096 and would thus enable uniform attention with Llama 2.
> >
> > I would still appreciate if you could incorporate some more of this discussion about precision into the paper, as well as some more explicit discussion of the non-uniform nature of your construction.
> >
> > Otherwise, I maintain my score.

---

> > > ### Author Response · Authors · 2023-11-23
> > >
> > > You are correct that the maximal number BF16 can express is $2^{2^8}$. But this does not mean BF16 can express every floating point number below the maximal number. In particular, BF16 cannot express any real number in the range of (256,258). Any number between 256 and 258 will be either rounded to 256 and 258. That means there will be a constant multiplicative error if one wants to use BF16 to express uniform attention over 257 elements.
> > >
> > > Per your request, we have updated the paper and included the discussion of precision in the related work section. We also added a section H in the appendix discussing non-uniformity. We explicitly mentioned our model is non-uniform in the main text of the revision.

---

### Official Review · Reviewer_zX88 · 2023-11-02

**Soundness:** 4 excellent
**Presentation:** 3 good
**Contribution:** 2 fair
**Rating:** 6
**Confidence:** 4

**Summary:**

This paper formally studies the representation power of transformers when used with chain-of-thought (CoT). The authors prove that transformers, under a certain formal model (but see weakness below of the model), with $T(n)$ CoT steps (where $T(n)$ is polynomial in $n$) on input of size $n$ can simulate the computation of any circuit of size $T(n)$. As a result, they derive a lower bound of P/poly. Along the way, they also derive new bounds on finite-precision and fixed-point log-precision transformers when rounding in iterated addition is done iteratively, rounding the sum of two numbers at a time.

**Strengths:**

The main strengths of this paper are:

1. its rigorousness in making clear, concise statements of its findings (except for one important "detail" discuss under weaknesses);

2. a formal characterization of the power of transformers with CoT, for which results have come out only very, very recently (after the ICLR submission deadline);

3. carefully crafted arguments and proofs around rounding of numbers when performing addition of $n$ numbers, a key step used in multiple prior papers unless less realistic assumptions; and

4. empirical evaluation to support the theory, which is often missing in similar theoretical characterizations of transformer variants in the past.

I have not read the proofs in detail (esp. the material in the appendix), but the results and approach intuitively appear plausible.

**Weaknesses:**

I really have only one, albeit big, concern about the paper, namely **the formal model of transformers being studied is different not only from all prior theoretical works but also from practical use of transformers**. This is made worse by the lack of a discussion of this difference. Consequently, while the results appear to tighten prior upper bounds and provide novel lower bounds, they really are applicable to a different model.

Specifically, the authors assume a model of transformers that is **non-uniform** (in the sense it is used in circuit complexity), namely, for each $n$, there is a **different** transformer. As they state in Defn 3.4, "for every ... $n$, there is a $L$-layer ... transformer". This means that one needs a *family* of transformers, one for each input size $n$, to solve a given problem, and the weights of the transformer for input length $n$ may have nothing to do with that of the transformer for input lengths $n+1$. To specify such a family, one thus needs to specify an infinite family of unrelated weights, which obviously is unrealistic.

In contrast, in practice, one trains a transformer on inputs of certain lengths, freezes those weights, and then uses the same frozen-weights transformers for inputs (and chains-of-thought) of arbitrary lengths. In fact, inspired by this, the theoretical research on the representation power of transformers in the past 2-3 years has gone in the opposite direction---from non-uniform upper bounds, to tighter and tigher uniform bounds (e.g., log-space uniform, log-time uniform, FO-uniform, etc.).  Importantly, all along, the model of transformers used was uniform.

The *non-uniformity* of the model assumed here has other, well-known undesirable consequences also seen in non-uniform circuits---it allows transformers to **trivially solve certain undecidable problems**, namely any undecidable unary problem, over the alphabet $\{1\}$. E.g., it can solve the halting problem expressed in unary, just like all non-uniform circuit classes (including P/poly) can.

Besides being unrealistic, this raises a question about the meaningfulness of the technique used to prove the main result (Theorem 3.3). In order to simulate a circuit of size $T(n)$, one must somehow *embed* the circuit into the transformer as the transformer needs to know what circuit to compute. As seen from the proof of it in the appendix, the authors have a creative solution: they put the description of the circuit in the *positional embedding* of the transformer.

While interesting and unique, this has two undesirable implications:

1. The positional embedding for inputs of length $n$ is allowed to be arbitrarily different from the positional embeding for inputs of length, say, $n+1$ (because there is no uniformity constraint between the circuits for the two respective sizes, $n$ and $n+1$), which departs heavily from practice.

2. It means that the proposed construction must include some **uncomputable / undecidable positional embeddings**! To see this, consider any undecidable unary problem $P$ in P/poly. The typical polysize circuit construction for $P$ is to have, for each $n$, a trivial circuit $C_n$ that outputs a $1$ if and only if $1^n$ is in that undecidable language (e.g., $1^n$ is the unary encoding of the $n$-th Halting problem). Thus, by the authors' construction, there is a trivial transformer that decides the same language---and its embedding of the first position computes membership in that undecidable language! In other words, the embedding itself in not computable by any reasonable model of computation.

To summarize this, while the construction is correct to my understanding, it is for a formal model that departs from practice and assumes a lot of power (e.g., that of having access to potentially uncomputable embeddings).

At the very least, these limitations and their implications should be clearly discussed in the paper.

**Minor points**

* In the abstract, the statement "with $T$ steps of CoT, constant-depth transformers ... can solve ..." should be qualified with T being at most a polynomial in $n$.

* In the 2nd last line of page one, I think you mean "encoder-only" rather than "decoder-only"; or single-step decoder.

* In line 3 of section 2, do you mean $\phi(bin_k(x)) = x$ rather than $\ldots = 0$?

* In the 2nd paragraph of section 2, where is the input length limited to $n_\max$?

* page 3, two lines before defn 3: "over more two" => "over more than two"

**Questions:**

1. Please see my main concern above regarding non-uniformity. I don't really have any specific question around it, though I wonder what you think about the non-uniformity issue above and how, if you choose to, would you incorporate it in a revised version of the paper.

2. After Theorem 3.2, you mention that an analog of this theorem remains open for log-precision transformers even with a constant number of bits for the exponent. Could you elaborate why one can't just absorb the constant bits of the exponent into additional constant bits of the significand? Perhaps you can illustrate it with what goes wrong if one has $1$ bit for the exponent?

---

> ### Author Response · Authors · 2023-11-22
>
> **Q1: Our computation model is non-uniform.**
>
> > **Response**: Thanks you for your this critical question and constructive feedbacks. Please refer to our general response on the non-uniformty issue to all the reviewers.
>
> **Q2: An analog of Theorem 3.2 remains open for log-precision transformers even with a constant number of bits for the exponent. Why cannot we absorb constant bits of exponent into additional constant bits of the significand?**
>
> > **Response**: A short answer is that the existing proof of $\mathsf{TC}^0$ upper bound relies on the fact that the addition among fixed point numbers satisfy the association rule (assuming no overflow happens), and floating point numbers with any non-zero bits for the exponent does not satisfy the association rule. Taking the floating point system defined in our manuscript as an example, let both the number of exponents and precision, namely, $e$ and $s$ be $1$, we have $\mathbb{F} _{1,1} = \{-3/2,-1,-3/4, -1/2,-1/4, 0,1/4,1/2,3/4,1,3/2\}$. One can note that $5/4$ and $-5/4$ does not belong to $\mathbb{F} _{1,1}$ due to the limited precision of $\mathbb{F} _{1,1}$. As a result, even though $a= -1,b=1,c=1/4$ all belong to $\mathbb{F} _{1,1}$, we have that $[[a+b] _{1,1}+c] _{1,1} \neq [a+[b+c] _{1,1}] _{1,1}$. This is because
> > $$ [[-1+1] _{1,1}+1/4] _{1,1} = [0+1/4] _{1,1} = 1/4\neq 0 = [-1+1] _{1,1} = [-1+[1+1/4] _{1,1}] _{1,1},$$
> > where the last is because we break the tie in rounding by picking the number with smaller absolute value (See Definition 3.2).

---

### Official Review · Reviewer_Aro9 · 2023-11-03

**Soundness:** 3 good
**Presentation:** 4 excellent
**Contribution:** 3 good
**Rating:** 8
**Confidence:** 2

**Summary:**

This paper investigates whether chain of thought(COT) increases the power of transformers. COT provides additional power to transformer decoders in the form of a hidden "scratch pad" where a transformer can perform serial computations. Without COT, a transformer is limited to being a fixed depth circuit, and as such is known to have very limited power in the complexity sense. The paper studies the maximum ability of a transformer given the ability to run COT. COT is able to significantly expand the ability of transformers as without COT transformers are in AC0 and with COT transformers can solve problems from a significantly larger class, all problems with solvable by Boolean circuits of size T (T= time steps given to COT). The key technical contribution is a formalisation of COT and its placement in the circuit complexity heirarchy.

**Strengths:**

Overall i felt it was a quite well written paper and the case was made well. The paper formalizes the problem precisely, which is not only critical for answering the question considered in the paper, but is also of independent interest. I think the parameterisation of CoT  in terms of embedding size and depth, is novel and interesting. There are several well chosen examples that made comprehension easier.

**Weaknesses:**

The empirical evaluation is fairly convincing but it does not really reveal anything new insights not already covered by the proofs.

Minor  errors:
Section 1: poewrful -> powerful

**Questions:**

--

---

> ### Author Response · Authors · 2023-11-22
> **Response to Reviewer Aro9**
>
> We thank you for your appreciation and your effort devoted to the review. We will fix the typo you pointed out in the revision.

---

### Official Review · Reviewer_pRpF · 2023-11-06

**Soundness:** 3 good
**Presentation:** 2 fair
**Contribution:** 4 excellent
**Rating:** 3
**Confidence:** 4

**Summary:**

This paper looks at the expressive power of transformer models, specifically looking at the expressive power on sequential reasoning tasks rather than parallel tasks. The main results are theoretical and It introduces several new classes of problems that are tightly focused

**Strengths:**

To my mind, there were three main strengths to the paper.

1. Its theoretical approach: I greatly appreciated the theoretical bend to the paper. Phrasing things in terms of problem classes that a particular model can solve is the kind of I'd like to see more of.

2. Interesting examples: The problem classes they used were interesting, novel, and nicely targeted to the theoretical results

3. A detailed and rigorous approach to a intuitive idea. The supporting information section that includes very detailed proofs and ample details for the interested reader.

In terms of the primary dimensions, this paper was quite strong in two of them: Originality and Significance

Originality: As previously noted, phrasing the CoT problem in terms of specific problem classes for particular families of models is a fantastic idea, something we don't see nearly enough of in the field. I also thought the experiments performed were quite original - finding problems in particular problem classes takes a great deal of effort and creativity, the problems they used were not brand new, but were new to the area of CoT and transformer models

Quality: The experiments performed were well chosen and did a good job supporting the theory

Significance: This paper does make a significant claim and provides good evidence to back it up. Understanding the power of transformer models, where they face challenges, and what problem classes they excel at, is a very significant result and exactly the kind of result that should be featured at an ICLR tier conference.

**Weaknesses:**

For all the strengths of this paper, there were a number of weaknesses as well.

The performance evaluation - data was trained and evaluated using "freshly sampled synthetic data". I think this is problematic at several levels. First, without a carefully though out test/train split, the model is at risk of overfitting. I think that is fine here - because overfitting is still telling you information about the expressive power of your model class, however this goes against the grain of standard model training practice, and at least deserves comment. Second, the problems used were all discrete problems. Discrete problems are great test beds for theoretical arguments - but in most of these examples, the problem space is finite for a fixed size, and randomly sampling from a discrete space gives the artificial impressing of having more data than is actually available. Finally, how you sample from these discrete spaces seems like it would make a big difference on the model performance

The model discussion was also significantly lacking. At the very least, a discussion of what the class of transformer models looks like and how you are bounding it belong in the main body of the text, not banished to appendix G on the final page of the supporting information. The details that were present, were in the form of Algorithm 1, defining an implementation, but not the class itsself. That made it difficult to tell what was the result of the model class and what was the result of the training process.

Finally, this paper would be stronger if the presentation was as crisp and focused as the data classes and results. In particular, there was an odd mix of too much detail, too little detail, and superfluous detail. For example, Definitions 3.1 and 3.2 were primarily used in appendix C so their presentation on page 3 distracted from the main arguments of the paper.

**Questions:**

What is the assumed background of a reader of this paper - as someone quite familiar with both machine learning, complexity theory, and the combinatorial problem classes being addressed, I found this paper confusing - for who I imagine the typical reader to be, I think lots of the content of this paper can be assumed knowledge - for example, I don't think a definition for a finite state automaton (Definition D.1) is a productive use of space in this paper - this is something you can cite directly (e.g. text above corollary 3.4 on p.6).

In all the figures, I did not find the presented examples particularly illuminating. For instance, with Figure 1, with the non CoT example,  the given label does not appear to be a composition of the permutations given in the input - and it doesn't even appear to have the form of the right answer.



Suggestions
- This paper spends a lot of time on the setup and has an extensive collection of supporting material. I like t

---

> ### Author Response · Authors · 2023-11-21
> **Response to Reviewer pRpF**
>
> We thank the reviewer for the thorough and constructive feedbacks.  We also sincerely appreciate the reviewer's recognition of the significant contribution made by our work. Below we address the concerns of the reviewer one by one and would respectfully ask the reviewer to consider increasing the score if our responses satisfacorily address the concerns.
>
> **Q1: Training setup not clear. Is there a risk of overfitting?**
>
> > **Reponse**: There is no risk of overfitting because in the experiments, we train in an online setting -- at each step, the training batch is i.i.d. sampled from distribution. We will highlight on this in the revision. Thanks for pointint this out!
>
> **Q2: Discrete problems have finite problem space**
>
> > **Reponse**: We agree with this point, but we do not agree that this is a weakness. Any language problem with finite vocabulary and finite sequence length has a finite problem space. We hope the reviewer could elaborate more on this critism.
>
> **Q3: Sampling method of the data distribution make a huge difference.**
>
> > **Reponse**: We agree. But to support our theoretical claim that there is a separation between expressiveness of transformers with and without chain of thought, it suffices to show one data distritbution.
>
> **Q4: The model details should not be hidden in appendix. The presentation can be more crips and focused.**
>
> > **Response (updated)**: Thanks for this helpful suggestion on the presentation. Following your suggestion, we moved the original Definitions 3.1 and 3.2 into the appendix and brought the definition of our transformer architecture into the main paper. See Section 2 in the revision. We also added a clarification in our paper that our model architecture is the same as GPT-2 [Radford et al., 2019], except we use ReLU instead of GeLU. This architecture is also used by GPT-3 with slight modification [Brown et al., 2020].
>
> **Q5: Expected background of readers**
>
> > **Reponse**: We hope both language model and computational complexity community will find this paper interesting. Given the potential diversity in readers' background and that ICLR has no page limit for the appendix, we choose to make our paper more self-contained by providing those basic definitions in the appendix, including both about decoder-only transformers and complexity.
>
> **Q6: The label in Figure 1 for the non CoT example seems wrong**
>
> > **Reponse**: It is indeed correct. Here the composition rule for permutation is that, given two permutation $\sigma = (\sigma_1,\ldots,\sigma_5)$, $\pi = (\pi_1,\ldots,\pi_5)$, we define $\sigma\circ \pi \triangleq (\sigma_{\pi_1},\ldots, \sigma_{\pi_5})$. Here in the example of Figure 1, we have $\sigma^{(1)} = (2,3,1,4,5)$, $\sigma^{(2)} = (1,2,4,3,5)$, $\sigma^{(3)} = (5,4,3,2,1)$. Thus for chain of thought, we have $\sigma^{(1)} \circ \sigma^{(2)} = (\sigma^{(1)} _{\sigma^{(2)} _1}, \ldots, \sigma^{(1)} _{\sigma^{(2)} _5}) = (\sigma^{(1)}_1, \sigma^{(1)}_2,\sigma^{(1)}_4, \sigma^{(1)}_3, \sigma^{(1)}_5) = (2,3,4,1,5)$.
>
> > Similarly, we have $\sigma^{(1)} \circ \sigma^{(2)} \circ \sigma^{(3)} = \left( \sigma^{(1)} \circ \sigma^{(2)} \right) \circ \sigma^{(3)}  = (2,3,4,1,5) \circ (5,4,3,2,1) = (5,1,4,3,2)$, which is the output for the example in Figure 1.  We will be more clear about the definition of the composition of permutation in the revision.
>
> **Q7: incomplete comments**: Additionally, we note that the following sentence in the review **"Suggestions: This paper spends a lot of time on the setup and has an extensive collection of supporting material. I like t"** is incomplete. We would like to hear the reviewer's suggestion and improve our paper based on it.
>
>
> **References:**
> - Radford, A., Wu, J., Child, R., Luan, D., Amodei, D. and Sutskever, I., 2019. Language models are unsupervised multitask learners. OpenAI blog, 1(8), p.9.
>
> - Brown, T., Mann, B., Ryder, N., Subbiah, M., Kaplan, J.D., Dhariwal, P., Neelakantan, A., Shyam, P., Sastry, G., Askell, A. and Agarwal, S., 2020. Language models are few-shot learners. Advances in neural information processing systems, 33, pp.1877-1901.

---

### Official Review · Reviewer_wS3v · 2023-11-06

**Soundness:** 3 good
**Presentation:** 3 good
**Contribution:** 3 good
**Rating:** 8
**Confidence:** 2

**Summary:**

The paper studies (mostly theoretically) the experessiveness of transformer architectures. The authors show that transformers with chain of thought can compute boolean circuits with a number of steps that is not bounded by the depth of the transformer.

**Strengths:**

(+) Important result regarding the expressiveness of transformers. The finding seems very obvious intuitively, but the novelty seems to be in a rigorous proof.

(+) Realistic modeling of finite precision computations

(+) The theoretical finding is matched with empirical results on four tasks in arithmetic.

**Weaknesses:**

(1) I'm not sure I grasp the significance of the finding. It seems obvious that any model can perform serial computations if it is allowed to store intermediate results somewhere (in this case, in the output sequence, which can be written to and subsequently access with self-attention). Therefore, producing such intermediate output seems as expressive as a chain of multiple instances of the model (this does not even specific to transformers).

Is this intuition misguided? Perhaps the important finding is rather that a transformer *without* CoT could not solve serial problems? (but this was covered in previous work).

Note that this topic is not exactly my area of expertise (I was called as an emergency reviewer), but this should be addressed since it is likely something that other readers will wonder about.

---------------------

(2) Presentation could be improved. These are details though that are easy to fix.

The abstract could be clearer about what form of CoT is studied in the paper (whether it's about the training data, the test phase, the prompting, ...). The very first sentence seemed clear in retrospect to me, not on the first read. So it's probably good to be even more explicit and specify that it's just about instructing the model to generate intermediate steps.

There are some informal shortcuts in the technical language that should be fixed. Examples:
"transformers with polynomial intermediate steps" -> "with a number of intermediate steps polynomial in ..."
"transformers with linear intermediate steps"
"poly width and log precision"
etc.

Typos: poewrful, circuit valuation, "because every regular languages" (should be singular)

Sec. 4"we find that cotis always" (missing space, also in multiple other places in this section)

Weird grammar in the abstract: "previous works have ..."
I suppose you mean "previous works show that ..."

**Questions:**

See (1) above.

---

> ### Author Response · Authors · 2023-11-22
> **Response to Reviewer wS3v**
>
> We thank the reviewer for the helpful feedback. We have corrected the notations
>
> **Q1: The main finding that transformers with chain of thought can do serial computation sounds obvious and is not significant.**
>
> > **Response**:  While we agree that the ability to perform serial computation might sound natural for computation models with large computation depth, a rigorous construction for a specific architecture, transformers with chain of thought, is novel and not trivial, especially given the various restrictions, like constant precision and $\log(n)$ embedding size. For example, previous work (Perez et al., 2021) also shows that encoder-decoder transformers can perform inherently serial computation by showing that transformer can use one step of chain-of-thought to simulate one step execution of Turing Machine. But this construction uses infinite precision and does not apply in our setting, where only constant precision is allowed.
>
> **References**:
> - Pérez, J., Barceló, P. and Marinkovic, J., 2021. Attention is turing complete. The Journal of Machine Learning Research, 22(1), pp.3463-3497.

---

### Official Review · Reviewer_qiA8 · 2023-11-09

**Soundness:** 3 good
**Presentation:** 3 good
**Contribution:** 3 good
**Rating:** 5
**Confidence:** 4

**Summary:**

- The purpose of this study is to demonstrate that CoTs can enhance the reasoning ability of LLMs. To prove this, they utilize the language of complex circuits to compare the expressive power of Transformer models with and without CoTs.

- First, they hypothesize that CoTs enable more efficient serial computations, which vanilla Transformers cannot achieve. they subsequently establish that, when dealing with sufficiently long inputs, the precision of a Transformer model is limited to solving problems within $AC^0$. Furthermore, they demonstrate that CoTs, with T steps, utilizing a constant-depth transformer, constant localization accuracy, and O(log n) embedding size, can solve problems that exceed the capabilities of Boolean circuits of size T. Finally, they find that increasing the embedding size to poly(n) does not improve expressiveness beyond using a log(n) embedding size.

- In conclusion, their theoretical analysis suggests that models with CoTs exhibit enhanced expressive abilities when compared to Decoder-only Transformers without CoTs. they have designed experiments to validate their theoretical findings.

**Strengths:**

- Analyzing the effect of the presence or absence of CoT on model expressivity from a theoretical perspective.
- Proposing a tighter upper bound for constant depth transformers' expressive power.
- The motivation of this paper is clear.

**Weaknesses:**

- Symbols are not clearly described. The authors  don't make clear instructions when using symbols like NC^1, AC^0, etc. It is recommended that these symbols be explained when they are first mentioned.

- The conclusions of Theorem 3.8 and Theorem 3.9 presented in the paper seem to be contradictory. Theorem 3.8 indicates that enlarging the embedding size to poly(n) doesn't improve the model's expressiveness of T(n) = poly(n) step CoT. On the other hand, Theorem 3.9 demonstrates that broadening the embedding width strengthens the model's power for any particular polynomial T(n) = n^k step CoT. The conclusions of these two theorems are confusing and require further clarification.

**Questions:**

- I would like the author to supplement and polish the article based on the weaknesses.
- In the demonstration experiment shown in Figure 2, it is observed that when the depth is set to 1, the CoT performs less effectively compared to the decoder-only Transformer without CoT. This difference is particularly noticeable when the label represents the sum of the inputs modulo 2 and exceeds a threshold of 20%. This observation appears to contradict the result presented in Theorem 3.2. Is related to the fact that CoT simulates any depth circuit by writing the output token back to the next input position? Further clarification from the author regarding this matter would be greatly appreciated.

---

> ### Author Response · Authors · 2023-11-22
> **Response to Reviewer qiA8**
>
> **Q1: Symbols are not clearly described. The authors don't make clear instructions when using symbols like NC^1, AC^0, etc. It is recommended that these symbols be explained when they are first mentioned.:**
>
> > **Response**: Due to the space limit, we defer the definition of complexity class to Appendix B, as noted in our section 2, notations and preliminaries.
>
> **Q2: The conclusions of Theorem 3.8 and Theorem 3.9 presented in the paper seem to be contradictory. Theorem 3.8 indicates that enlarging the embedding size to poly(n) doesn't improve the model's expressiveness of T(n) = poly(n) step CoT. On the other hand, Theorem 3.9 demonstrates that broadening the embedding width strengthens the model's power for any particular polynomial T(n) = n^k step CoT.**
>
> > **Reponse**: There is no contradiction. We would like to clarify that Theorem 3.8 does not say that "enlarging the embedding size to poly(n) doesn't improve the model's expressiveness of T(n) step CoT **for any fixed polynomial T(n)**". Instead, Theorem 3.8 should be interpreted as "for every polynomial T(n) and every function expressible by poly embedding size transformers with T(n) embedding size, there is a different and potentially larger polynomial function T'(n) than T(n), and another transformer with log-embedding size and T'(n) steps of COT, which can also express the function".
>
> **Q3:  In Figure 2, it is observed that when the depth is set to 1, the CoT performs less effectively compared to the decoder-only Transformer without CoT. This difference is particularly noticeable when the label represents the sum of the inputs modulo 2 and exceeds a threshold of 20%.**
>
> > **Response**: Thanks for pointing this out. We think this is because the modular addition experiments are the only set of experiments which we do not exactly follow the setting of our theorems. Because it is so easy to achieve perfect accuracy, we use a more challenging training task there, that is, the data distribution contains the input of different sequence length. The idea is that if the models can achieve perfect accuracy on such data distribution, then it shows it can express the ground truth of modular addition for each sequence length. The benefit of doing this is that we can save space for the plot.
> >
> > However, the above deviation from the theoretical setting could indeed make cot not as powerful as hint when the depth is low, as it is beyond the theoretical guarantee. To avoid confusion, we rerun the experiments for the modular addition using data distribution of fixed sequence length, and the results are presented in Figure 2, Figure 5, and Figure 7 in the revision. In the new experiments, we can clearly see cot is always better than hint, and hint is always better than base.
> >
> > Due to the time constraints, we only rerun the experiments for depth equal to 1, 2, and 3 and we will add the remaining depth up to 12 in the future revision. Thank you again for this critical and constructive feedback!

---

### Author Response · Authors · 2023-11-22
**Response to concerns on non-uniform embeddings**

Reviewer zX88 and woBt raised the questions about non-uniformity of our complexity class. We would like to address these concerns below.

1. **Is non-uniformity a fundamental property towards showing the expressiveness lower bound of the transformers?**

> **Reponse**: No, it is really more about convenience. We can add an additional assumption that the weights of the transformers must be able to be generalized by a uniform $\mathsf{AC}^0$ given the total sequence length $n$. The expressiveness lower bounds in the paper still hold after this modification, but now they become uniform circuit complexity class. For example, $\mathsf{P/poly}$ becomes $\mathsf{P}$. This is because of the standard result that one can generate the boolean circuit of size $O(T^2)$ to simulate the running of Turing Machine for $T$ steps.

> Similarly, boolean circuits are inherently non-uniform as a computation model as one needs to use a different circuit for every different input length. But they are still among the most popular computation models in computation complexity.

2. **Having a different transformer for each input sequence length is different from the practical use of transformers.** (Reviewer zX88)

> **Reponse**: We disagree. In practice when dealing with longer input sequence length, the embedding size and depth of the transformers typically increase as well. To our best knowledge, there is no single language model that can handle any input sequence length.

> Moreover, the architecture in our paper is directly copy-and-pasted from GPT-2 architecture [Radford et al.,2019], except for minor changes like changing GeLU to ReLU. This architecture is also inherited by GPT-3 with slight modification on attention. [Brown et al.,2020] It is unfair to claim the the transformer model studies in our paper is impractical.


3. **Having a different transformer for each input sequence length is different from the previous theoretical works and thus the tightened upper and new lower bound are not comparable.** (Reviewer zX88)

> **Reponse**: We disagree.

> - For lower bounds, the most related work to ours is [Liu et al., 2022] where they show that constant depth transformers can compute solvable automata and log depth transformers can compute all automata. This result is non-uniform and also uses absolute position embedding. Our lower bound shows that with COT of linear length, constant depth transformers are enough to compute all automata, thus demonstrating the benefit of CoT in expressiveness.

> - Most previous **upper bounds**, like AC0 upper bounds for constant depth transformers with hard attention [Hao et al., 2022], and TC0 upper bounds for constant depth transformers [Merrill et al.,2022, Liu et al.,2022, Merrill and Sabharwal, 2023], hold for non-uniform transformer class as well. Our new $\mathsf{AC}^0$ upper bound is comparable to those results. Note that all these lower bounds do not hold for transformer using floating numbers with non-zero exponent bits because of lack of association rule addition (see a concrete example in the response to Reviewer zX88, Q2), our new $\mathsf{AC}^0$ upper bound for constant precision transformer is novel in two senses: first, it is tighter than best previous known upper bound for $\mathsf{TC}^0$, and second it is the first upper bound truly floating-point arithmetic, instead of fixed-point arithmetic, i.e., floating-point number with no exponent bits.
> That being said, we agree that our upper bounds are not comparable with those uniform results, like Theorem 2 in [Merrill and Sabharwal, 2023]. We will clarify this in the revision. Uniform results are in general more difficult, and for standard attention function, the biggest issue seems to be how to even compute the correct rounding of exponential of $\log(n)$ precision floating numbers in $\mathsf{poly(n)}$ time. To our best knowledge, this question is open and the solution in literature is typically treat the exponential function as a non-uniform circuit with $\mathsf{poly(n)}$ size, e.g., [Liu et al.,2022]. Indeed, the current way of computing exponential function in IEEE 754 is inherently non-uniform, where one uses a different algorithm for every precision, e.g., see [Defour et al., 2001]. Of course, one way to circumvent is to consider an approximation of an exponential function that is easy to compute, like the truncation of Taylor series of the exponential function. But this would make the analysis less realistic.

We hope the above answers address all the current concerns of the reviewers. However, we also feel like maybe the real question at reviewers' heart is learnability.

---

> ### Author Response · Authors · 2023-11-22
> **Response to concerns on non-uniform embeddings (continued)**
>
> 4. **Can transformer architectures with more trainable parameters when increasing input sequence length achieve length generalization? That is, train on shorter lengths, and generalize to unseen longer lengths.**
>
> > **Response:** Honestly we don't believe the answer to the above question is positive. It is well-known that absolute position embedding can lead to the failure of length generalization. However, this does not stop the success of GPT architectures, which use trainable absolute position encoding. The explanation behind this is that in the real world, language models still have a limit for input length. Moreover, for longer input sequences, people also tend to use models with large embedding sizes. In other words, there is some disconnection between practice and the theoretical value of uniformity. We do not think we can decide which regime is correct without further advancement in practice. That being said, we do agree it sounds impossible to achieve length generalization with additional trainable parameters for longer input length.
>
>
> **References:**
>
> - Hao, Y., Angluin, D. and Frank, R., 2022. Formal language recognition by hard attention transformers: Perspectives from circuit complexity. Transactions of the Association for Computational Linguistics, 10, pp.800-810.
>
> - Merrill, W., Sabharwal, A. and Smith, N.A., 2022. Saturated transformers are constant-depth threshold circuits. Transactions of the Association for Computational Linguistics, 10, pp.843-856.
>
> - Liu, B., Ash, J.T., Goel, S., Krishnamurthy, A. and Zhang, C., 2022. Transformers learn shortcuts to automata. arXiv preprint arXiv:2210.10749.
>
> - Merrill, W. and Sabharwal, A., 2023. The parallelism tradeoff: Limitations of log-precision transformers. Transactions of the Association for Computational Linguistics, 11, pp.531-545.
>
> - Defour, D., De Dinechin, F. and Muller, J.M., 2001, November. Correctly rounded exponential function in double-precision arithmetic. In Advanced Signal Processing Algorithms, Architectures, and Implementations XI (Vol. 4474, pp. 156-167). SPIE.

---

### Author Response · Authors · 2023-11-23
**Updates in Revision**

In the rebuttal revision, we made the following updates, which are marked in red in the revision.

1. We added a section H in the appendix discussing non-uniformity. We also explicitly mention our model is non-uniform in the main paper. (in request of reviewer **zX88** and **woBt**)

2. We reran the experiments of modular addition so that now it exactly follows the setting of theoretical analysis, like the rest experiments in the paper. See Figure 2, 5, and 7. This addresses the concern of Reviewer **qiA8**.

3. We added a paragraph discussing the issue of logarithmic precision versus constant precision in related works. (in request of Reviewer **woBt**)

4. We cited all the related works mentioned by the reviewers.

5. We fixed typos and improved writing as suggested by the reviewers.

---

### Meta-Review · Area_Chair_2Fe5 · 2023-12-11

**Metareview:**

This paper was reviewed by six experts with mixed reviews.  AC does feel that this work makes interesting contributions. The reviewers did raise some valuable concerns. The authors are encouraged to make the necessary changes and include the missing references in the final version.

**Justification For Why Not Higher Score:**

The experiment could be more comprehensive.

**Justification For Why Not Lower Score:**

NA

---

### Decision · Program_Chairs · 2024-01-16

Accept (poster)